# HOPFORMER: HOMOGENEITY-PURSUIT TRANS­FORMER FOR TIME SERIES FORECASTING

## ABSTRACT

Forecasting multiple time-series with high-dimensional covariates presents a core challenge: unifying common temporal patterns while retaining meaningful series-specific information. We introduce Hopformer (**Ho**mogeneity-**P**ursuit Trans**former**), a two-stage forecasting framework that addresses this challenge. In the first stage, our Sparsity Pattern Aggregation (SPA) scheme extracts a common, low-variance trend incorporating the covariates. This acts as a homogenization layer. In the second stage, a LoRA-fine-tuned Transformer models the remaining complex dependencies in the residual signals. Our method is theoretically grounded. We prove that SPA achieves a near-optimal bias-variance trade-off via an oracle inequality. We also provide generalization bounds for the second stage under dependent time series data. Hopformer sets a new state of the art, improving the MASE by an average of 6.56% on both synthetic and real-world benchmarks.

## 1 INTRODUCTION

Deep learning, particularly Transformer architectures, now sets the state of the art in time series forecasting, capturing complex nonlinear dynamics that challenge traditional models (Zhou et al., 2021; Li et al., 2019; Zhou et al., 2022; Zhang & Yan, 2023; Wu et al., 2021). Yet, these models often struggle when predictions depend on high-dimensional external covariates such as economic indicators or sensor arrays, as traditional attention or embedding layers in Transformers become computationally expensive or unstable. The newest generation of universal forecasting models only deepens this problem (Woo et al., 2024; Liu et al., 2024a; Ansari et al., 2024; Liu et al., 2024c; Das et al., 2024). While achieving impressive zero-shot generalization, they typically sidestep the covariate challenge due to architectural or scalability constraints. For instance, MOIRAI (Woo et al., 2024) supports covariates via its any-variate attention mechanism but has limitations for high-dimensional covariates, while Chronos (Ansari et al., 2024) discards them entirely. This leaves a critical gap: How can we harness the rich information in high-dimensional covariates to improve forecasting, without compromising the scalability and power of modern pretrained backbones?

To bridge this gap, we propose Hopformer (**Ho**mogeneity-**P**ursuit Trans**former**), a novel two-stage forecasting framework designed to unify common temporal dynamics across high-dimensional, multi-source time series. Our central insight is to decompose the forecasting task into two complementary modules—(1) deterministic trend extraction that captures low-frequency, covariate-driven structure, and (2) residual modeling that handles nonlinear and long-range temporal dependencies. This separation enables Hopformer to integrate high-dimensional covariates while remaining modular and scalable, and to improve foundation models with minimal re-training. The two-stage design is supported by theoretical guarantees for both components. In the first stage, Hopformer extracts shared trend signals from covariates using a pool of cross-sectional regression experts, including linear, tree-based, and neural models. These experts are adaptively aggregated using *Sparsity Pattern Aggregation* (SPA) (Rigollet & Tsybakov, 2011), a convex model combination scheme that balances empirical risk with model complexity. This stage serves as a homogeneity-pursuit interface: it aligns heterogeneous series by projecting high-dimensional covariates into a lower-dimensional shared trend space. We provide an oracle inequality for this estimator, showing that SPA yields near-optimal prediction error under mild sparsity assumptions. In the second stage, Hopformer models residuals using a pre-trained Transformer fine-tuned via *Low-Rank Adaptation* (LoRA) (Hu et al., 2022), a parameter-efficient tuning method. We adopt LoRA not only for its empirical efficiency, but also because it enables generalization guarantees under time series dependence: building on information-theoretic arguments, our theory shows that fine-tuning reduces the mutual information between model

and data. Our experiments show that it outperforms both zero-shot and full fine-tuning. The second stage is modular and can be replaced with other residual models depending on application needs.

This work addresses a fundamental challenge in time series forecasting: how to model high-dimensional covariates while capturing both structural trends and dynamic residuals. We propose a unified two-stage framework, Hopformer, that first extracts trends by aligning covariates and outcomes into a common representation space via expert-based regression, and then models residual variation using a fine-tuned Transformer. Our method introduces a sparsity pattern aggregation strategy for trend learning and provides theoretical guarantees at both stages: an oracle inequality showing SPA achieves near-optimal predictive error relative to the best expert subset, and generalization bounds for LoRA-based residual modeling under dependent data, offering insight into general PEFT.

## 2 RELATED WORK

**Universal Time Series Forecasting** Traditional forecasting methods often adopt a one-model-per-dataset paradigm, limiting scalability across heterogeneous time series (Wu et al., 2023; Nie et al., 2023; Liu et al., 2024b). To address this, recent advances leverage generative pretraining and prompt-based Transformers (Cao et al., 2024; Xue & Salim, 2023; Ekambaram et al., 2024; Jin et al., 2024), but often require custom modules for each task. In contrast, universal forecasting frameworks aim for broad generalization across tasks. SimpleTS (Yao et al., 2023) selects optimal models via time series type classification, while MOIRAI (Woo et al., 2024) and UniTS (Gao et al., 2024) integrate predictive and generative capabilities via enhanced Transformer encoders. FlexTSF (Xiao et al., 2024) further extends universality to irregular time series. These methods underscore a growing interest in flexible, general purpose time series forecasting foundation models.

**Expert Aggregation and Sparse Ensembles** Combining diverse regressors is a time-tested strategy to improve robustness in forecasting. Classical approaches include stacking (Godahewa et al., 2023) and mixture-of-experts (MoE) models (Jacobs et al., 1991), which dynamically assign weights to base learners. N-BEATS-MOE (Matos et al., 2025) extends this to temporal settings, improving adaptation to heterogeneous dynamics. On the theoretical side, sparsity pattern aggregation (SPA) (Rigollet & Tsybakov, 2011) offers a principled way to combine experts via exponential weighting, balancing empirical risk and model complexity. Follow-up work explores PAC-Bayesian guarantees (Dalalyan & Tsybakov, 2008) and affine estimator ensembles (Dai et al., 2014). Our work is the first to embed SPA into a two-stage temporal modeling pipeline, serving as a trend extractor before residual modeling.

**Parameter-Efficient Fine-Tuning for Time Series** Parameter-Efficient Fine-Tuning (PEFT) methods have gained popularity for adapting large pretrained models with minimal computational overhead. Techniques such as Low-Rank Adaptation (LoRA) (Hu et al., 2022), adapters (Houlsby et al., 2019), and prompt tuning (Lester et al., 2021) enable efficient fine-tuning by updating a small subset of parameters. These approaches have been successfully applied in time series domains (Gupta et al., 2024; Rasul et al., 2024; Ansari et al., 2024), demonstrating generalization across modalities and domains. In this work, we adopt LoRA as a representative PEFT technique due to its simplicity and compatibility with Transformer architectures. Our theoretical results provide generalization guarantees for this residual modeling stage, which can be extended to other PEFT methods.

## 3 METHOD

**Problem Formulation** We consider a dataset of $N$ time series $\mathcal{D} = \{\mathbf{X}^{(i)}\}_{i=1}^{N}$, where each time series $\mathbf{X}^{(i)} = (X_1^{(i)}, X_2^{(i)}, \ldots, X_{T_i}^{(i)}) \in \mathbb{R}^{T_i \times D}$ consists of $T_i$ time steps. Each time series shares a common set of covariates $\mathbf{Z}^{(i)} = (Z_1^{(i)}, Z_2^{(i)}, \ldots, Z_{T_i}^{(i)}) \in \mathbb{R}^{T_i \times d}$, where each column represents the same set of features including lagged values, seasonal terms, or trend indicators across all series at time $t$. While the target values $\mathbf{X}^{(i)}$ vary across time series, the associated predictors are constructed using a shared feature design, enabling consistent modeling and aggregation across heterogeneous time series. The objective is to develop a forecaster $F$ such that $\mathbf{X}_{t:t+h}^{(i)} = F(\mathbf{Z}_{t-l:t+h}^{(i)})$ for each $i$ and $t$, aiming to generalize across both short-term and long-term patterns. To address this, we adopt a two-stage framework designed to separate deterministic trends from residual dynamics. In the first stage, we use multiple expert models $\{g_j\}_{j=1}^{M}$ to extract trend components via cross-sectional regression: $X_t^{(i),\text{trend}} = \sum_{j=1}^{M} \omega_j g_j(\mathbf{Z}_t^{(i)}) + \epsilon_t^{(i)}$, where $\omega_j$ are aggregation weights learned through a sparsity pattern aggregation (SPA) scheme. This shared aggregation mechanism enables homogeneity pursuit across diverse time series while retaining adaptivity. In the second stage, we use a LoRA-fine-tuned Transformer model to further approximate the residuals $\epsilon_t^{(i)}$ with $\hat{R}_{t:t+h}^{(i)} = f_{\mathbf{W}}^{\text{LoRA}}(R_{t-l:t}^{(i)})$. While

LoRA is not strictly necessary, it aligns with our theoretical analysis on generalization bounds under dependent data and can be replaced by other PEFT methods. We finally aggregate the prediction as $\hat{X}_{t:t+h}^{(i)} = \hat{X}_{t:t+h}^{(i),\text{trend}} + \hat{R}_{t:t+h}^{(i)}$.

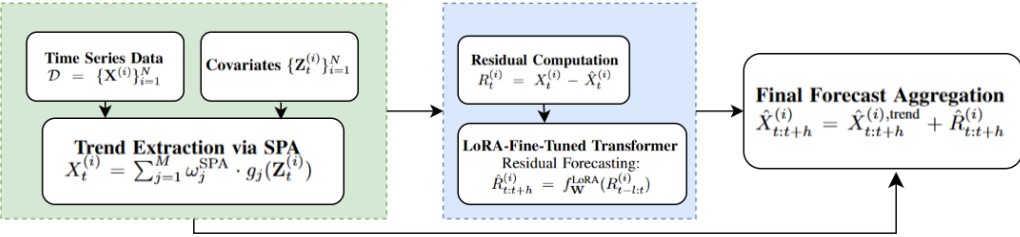

Figure 1: Overview of the Hopformer workflow. In Stage I, SPA extracts a low-variance common trend from covariates. In Stage II, a LoRA-fine-tuned Transformer models the residuals signal.

Universal time series forecasting remains challenging due to significant heterogeneity in temporal patterns, covariate structures, and data quality across datasets. While some methods explicitly incorporate covariates to improve generalization, such features are often sparse or inconsistently available—limiting their practical benefit. Instead of relying on manual feature alignment or domain-specific preprocessing, we propose a unified trend extraction strategy based on expert aggregation. Specifically, we construct a diverse pool of regression models—each capturing different structural aspects such as trend, seasonality, or autoregression—and combine them through a sparsity pattern aggregation (SPA) scheme. This approach acts as a homogeneity pursuit mechanism: by distilling shared deterministic signals from heterogeneous series, we project them into a common residual space where temporal dependencies are more stable. This decomposition confers several benefits: (1) it enables interpretable and flexible trend modeling without assuming a fixed parametric form; (2) it alleviates the need for hand-engineered covariates or alignment heuristics; and (3) it provides a interface for residual learning, improving the efficiency and stability of downstream models.

### 3.1 CROSS-SECTIONAL REGRESSION MODEL FOR TREND EXTRACTION

To extract the trend, we construct an ensemble of cross-sectional regression models $g_j{}_{j=1}^M$, each trained to capture different structural aspects of the time series using a shared covariate design. The aggregated trend estimate is given by a weighted combination: $X_t^{(i)} = \sum_{j=1}^M \omega_j^{\text{SPA}} \cdot g_j(\mathbf{Z}_t^{(i)})$, where $\mathbf{Z}_t^{(i)} \in \mathbb{R}^d$ denotes the covariates for time series $i$ at time $t$. Common features include lagged values, seasonal indicators, linear trend terms, and external covariates. For example, for a retail sales series, $\mathbf{Z}_t$ might include: a linear trend term ($t$); weekly seasonality terms; a binary promotion indicator $\text{Promo}_t \in \{0,1\}$; and a lagged value $X_{t-7}$ to account for weekly autocorrelation. To assign weights $\omega^{\text{SPA}} = (\omega_1^{\text{SPA}}, \ldots, \omega_M^{\text{SPA}})$, we employ *sparsity pattern aggregate* (SPA), which adaptively combines models based on empirical performance. While the optimal weights technically depend on both $t$ and $i$, we omit these indices for notational simplicity in what follows across this paper.

We define a sparsity pattern as a binary vector $\boldsymbol{p} \in \mathcal{P} := \{0,1\}^M$, where $p_j = 1$ indicates inclusion of model $g_j$ and $p_j = 0$ its exclusion. This pattern determines a restricted subspace: $\mathbb{R}^{\boldsymbol{p}} = \{\omega \cdot \boldsymbol{p} : \omega \in \mathbb{R}^M\} \subseteq \mathbb{R}^M$, where $\cdot$ denotes the Hadamard product. The dimensionality of this subspace is denoted by $|\boldsymbol{p}|$, the number of nonzero components in the pattern $\boldsymbol{p}$. To formalize the SPA construction, we consider the sample version of the workflow in Figure 1. For each time series $i$ and time point $t$, we assume access to an i.i.d. sample of size $n$ drawn from the underlying conditional distribution. Specifically, let $\{(X_{t,k}^{(i)}, \mathbf{Z}_{t,k}^{(i)})\}_{k=1}^n$ denote the observed realizations of the response and corresponding predictor vectors, where $\mathbf{Z}_{t,k}^{(i)} \in \mathbb{R}^d$ is the feature vector and $X_{t,k}^{(i)} \in \mathbb{R}$ is the target time series. Denote the sample version of target time series and the matrix of features as $\mathbf{X}$ and $\mathbf{Z}$. For any $\boldsymbol{p} \in \mathcal{P}$, let $\hat{\omega}_{\boldsymbol{p}}$ be the least squares estimator restricted to $\mathbb{R}^{\boldsymbol{p}}$, defined by $\hat{\omega}_{\boldsymbol{p}} \in \arg\min_{\pi \in \mathbb{R}^{\boldsymbol{p}}} \|\mathbf{X} - \mathbf{Z}\pi\|_2^2$. Let $\pi = (\pi_{\boldsymbol{p}})_{\boldsymbol{p} \in \mathcal{P}}$ be a probability measure over the collection of sparsity patterns $\mathcal{P}$, which referred as a prior over expert selections. The SPA weights are defined as:

$$\omega^{\mathrm{SPA}} := \frac{\sum\limits_{\boldsymbol{p} \in \mathcal{P}} \hat{\omega}_{\boldsymbol{p}} \exp\left(-\frac{1}{4\sigma^2} \sum\limits_{k=1}^{n} \left(X_{t,k}^{(i)} - g_{\hat{\omega}_{\boldsymbol{p}}}(\mathbf{Z}_{t,k}^{(i)})\right)^2 - \frac{|\boldsymbol{p}|}{2}\right) \pi_{\boldsymbol{p}}}{\sum\limits_{\boldsymbol{p} \in \mathcal{P}} \exp\left(-\frac{1}{4\sigma^2} \sum\limits_{k=1}^{n} \left(X_{t,k}^{(i)} - g_{\hat{\omega}_{\boldsymbol{p}}}(\mathbf{Z}_{t,k}^{(i)})\right)^2 - \frac{|\boldsymbol{p}|}{2}\right) \pi_{\boldsymbol{p}}}. \tag{1}$$

To instantiate the SPA weights, we adopt the following prior over the space of patterns $\boldsymbol{p} \in \mathcal{P}$ as in Rigollet & Tsybakov (2011): $\pi_{\boldsymbol{p}} := \frac{1}{H} \left(\frac{|\boldsymbol{p}|}{2eM}\right)^{|\boldsymbol{p}|}$ if $|\boldsymbol{p}| < R$; $\frac{1}{2}$ if $|\boldsymbol{p}| = M$; and $0$ otherwise. Here, $R = \mathrm{rk}(\mathbf{Z})$ is the rank of the design matrix, and $H = 2\sum_{m=0}^{R} \binom{M}{m} \left(\frac{m}{2eM}\right)^m$ is a normalization constant. This prior favors sparse patterns while ensuring that the full model retains sufficient mass for statistical guarantees and approximation stability. Following Rigollet & Tsybakov (2011), the SPA estimator under this prior can be efficiently implemented using a Metropolis approximation. This formulation allows Hopformer to adaptively emphasize simpler, performant trend models, serving as a learnable interface for residual modeling in the second stage. The residual sequence is then $R_t^{(i)} = X_t^{(i)} - \hat{X}_t^{(i)}$, contain long-range variation to be modeled in the second stage.

### 3.2 Fine-Tuning Transformer for Residual Forecasting

The residual sequence $R_t^{(i)}$ obtained from the trend extraction stage often contains complex, nonlinear, and long-range temporal dependencies that can be hardly captured by the cross-sectional regression ensemble. To model these patterns, we employ a Transformer architecture, which has demonstrated flexibility in modeling time series data due to its self-attention mechanism.

For a given residual input $R_{t-l:t}^{(i)}$, the Transformer models temporal dependencies by applying multi-head self-attention over learned representations. Specifically, the attention mechanism operates via query, key, and value projections:

$$\mathrm{Attn}(\mathbf{x}\mathbf{W}_q, \mathbf{C}\mathbf{W}_k, \mathbf{C}\mathbf{W}_v) = \mathrm{softmax}\left(\frac{\mathbf{x}\mathbf{W}_q(\mathbf{W}_k)^\top \mathbf{C}^\top}{\sqrt{d_k}}\right)\mathbf{C}\mathbf{W}_v, \tag{2}$$

where $\mathbf{W}_q^i, \mathbf{W}_k^i, \mathbf{W}_v^i \in \mathbb{R}^{d_{in} \times d_{out}}$ are learnable parameters and $\mathbf{C}$ denotes the context matrix.

To efficiently fine-tune the Transformer for each dataset while maintaining generalization, we adopt LoRA (Hu et al., 2022), which introduces trainable low-rank updates into the attention weights: $\mathbf{W} + \Delta\mathbf{W} = \mathbf{W} + \mathbf{B}\mathbf{A}$. Here, $\mathbf{W}$ is the original pre-trained weight matrix (kept frozen), and $\mathbf{B} \in \mathbb{R}^{d_{\mathrm{in}} \times r}$, $\mathbf{A} \in \mathbb{R}^{r \times d_{\mathrm{out}}}$ are trainable low-rank matrices with $r \ll \min(d_{\mathrm{in}}, d_{\mathrm{out}})$. The final residual update is applied as $\mathbf{h} \leftarrow \mathbf{h} + s\Delta\mathbf{h}$, $\Delta h := \mathbf{B}\mathbf{A}\mathbf{x}$, where $s \geq 1$ is a tunable scaling hyperparameter controlling the adaptation strength. This formulation allows fast adaptation with minimal memory footprint while preserving the expressive power. The residual forecasting function is thus expressed as $\hat{R}_{t:t+h}^{(i)} = f_{\mathbf{W}}^{\mathrm{LoRA}}(R_{t-l:t}^{(i)})$, where $f_{\mathbf{W}}^{\mathrm{LoRA}}$ is the LoRA-enhanced Transformer model. Then the final forecast combines the trend and residual predictions:

$$\hat{X}_{t:t+h}^{(i)} = \hat{X}_{t:t+h}^{(i),\mathrm{trend}} + \hat{R}_{t:t+h}^{(i)} \tag{3}$$

This two-stage strategy enables efficient and scalable forecasting, capturing both deterministic trends incorporating high-dimensional covariates and complex residual patterns across diverse datasets.

## 4 Theoretical Guarantees

We establish theoretical results for both stages. The first part analyzes the trend aggregation via an oracle inequality, while the second provides generalization bounds for the Transformer fine-tuning.

### 4.1 Oracle Inequality for Trend Aggregation

We analyze the performance of the trend aggregation procedure in the first stage of Hopformer, where the goal is to estimate an unknown regression function $\eta : \mathbb{R}^d \to \mathbb{R}^D$ based on observed data $\{(\mathbf{X}_t^{(i)}, \mathbf{Z}_t^{(i)})\}_{i=1}^N$. We assume the data is generated as $\mathbf{X}_t^{(i)} = \eta(\mathbf{Z}_t^{(i)}) + \xi_i$, $i = 1, \ldots, N$, where $\{\mathbf{Z}_t^{(i)}\}_{i=1}^N \subset \mathbb{R}^d$ are the covariates at time $t$ and $\xi_i \stackrel{\mathrm{i.i.d.}}{\sim} \mathcal{N}(0, \sigma^2 I_D)$, where $I_D$ is identity matrix, are independent Gaussian noise variables. Let $\hat{g}_\omega$ be the aggregated predictor using any weights $\omega$, and define the SPA estimator as $\hat{g}_{\mathrm{SPA}}(x) = \sum_{j=1}^M \omega_j^{\mathrm{SPA}} \cdot g_j(x)$, where $g_j$ is the $j$-th expert model. Then the following oracle inequality demonstrates the expected risk of the SPA estimator is near-optimal.

**Theorem 1.** *Under the model defined above, the SPA estimator $\hat{g}_{\mathrm{SPA}}$ satisfies the following inequality:*

$$\mathbb{E}\left\|\hat{g}_{\mathrm{SPA}} - \eta\right\|^2 \leq \min_{\boldsymbol{p} \in \mathcal{P}: \pi_{\boldsymbol{p}} \neq 0} \left\{ \mathbb{E}\left\|\hat{g}_{\hat{\omega}_{\boldsymbol{p}}} - \eta\right\|^2 + \frac{4\sigma^2 \log\left(\pi_{\boldsymbol{p}}^{-1}\right)}{n} \right\}, \tag{4}$$

*where $\pi$ is the prior distribution over sparsity patterns, $\hat{\omega}_{\boldsymbol{p}}$ is the least-squares solution restricted to pattern $\boldsymbol{p}$, and $\hat{g}_{\hat{\omega}_{\boldsymbol{p}}}(x) = \sum_j (\hat{\omega}_{\boldsymbol{p}})_j \cdot g_j(x)$ is the corresponding oracle predictor used in the SPA.*

The proof of this result is in Appendix A.1. This result ensures that the SPA estimator performs nearly as well—in expectation—as the best sparse combination of experts, with an additional complexity term depending on the prior mass $\pi_{\boldsymbol{p}}$ and the sample size $n$. The bound reflects a classic bias–variance trade-off: (1) The first term, $\mathbb{E}\left\|\hat{g}_{\hat{\omega}_{\boldsymbol{p}}} - \eta\right\|^2$, captures the approximation error from using the best subset of experts under pattern $\boldsymbol{p}$. (2) The second term, $\frac{4\sigma^2 \log\left(\pi_{\boldsymbol{p}}^{-1}\right)}{n}$, acts as a regularization penalty that grows with the model complexity (as encoded by the prior) and shrinks with more data. Intuitively, this inequality guarantees that when the prior assigns reasonable mass to good sparse patterns, the SPA procedure can adaptively discover and aggregate the most relevant components.

In the context of Hopformer, this result justifies our homogeneity pursuit strategy: by projecting diverse time series through an adaptively aggregated trend model, we obtain a denoised and compressed representation that retains meaningful structure across domains. This residualized representation becomes a more stable and generalizable input for the second-stage modeling by Transformer.

## 4.2 GENERALIZATION BOUND FOR LoRA FINE-TUNING

LoRA fine-tuning enables efficient adaptation of Transformer models to new time series data while keeping the number of trainable parameters low. To assess the reliability of this fine-tuning step in our residual forecasting pipeline, we establish a generalization error bound that holds under mild dependence assumptions typical in theoretical analysis under time series settings.

Let $\mathcal{Z}$ denote the space of residuals obtained from the first-stage regression. Suppose we observe a sequence $\boldsymbol{R}_T = (R_1, \ldots, R_T) \in \mathcal{Z}^T$ generated from a stationary process $\mathcal{P}$, and let $\mathcal{W}$ be the hypothesis space for model parameters. A learning algorithm produces a randomized predictor $\mathbf{W} \in \mathcal{W}$ based on $\boldsymbol{R}_T$, drawn from a conditional distribution $P_{\mathbf{W}|\boldsymbol{R}_T}$. Define the population risk and empirical risk as key metrics for evaluating model generalization:

$$L_{\mathcal{P}}(w) = \mathbb{E}_{\mathcal{P}}[\ell(w, R)], \quad L_{\boldsymbol{R}_T}(w) = \frac{1}{T-d} \sum_{i=d+1}^{N} \ell(w, R_i),$$

where $\ell : \mathcal{W} \times \mathcal{Z} \to \mathbb{R}^+$ is a loss function, and $d \geq 0$ is a burn-in offset. For example, in an AR(2) model, predictions depend on the past two time steps, meaning that at least two observations are required before a valid loss computed. Then the following theorem states a generalization bound for empirical risk using the information-theoretic framework.

**Theorem 2.** *Assuming the loss function $\ell(w, R)$ is $\sigma$-subgaussian. If the target residual time series is stationary and $\beta$-mixing, then there exists a constant $a > 0$ and an integer $m$ such that $2am \leq T$ ensuring*

$$\mathbb{E}[L_{\mathcal{P}}(\mathbf{W}) - L_{\boldsymbol{R}_T}(\mathbf{W})] \leq \sqrt{2\sigma^2 m^{-1} I(\boldsymbol{R}_T; \mathbf{W})}, \tag{5}$$

*where $I(\boldsymbol{R}_T; \mathbf{W}) = D_{KL}(P_{\mathbf{W}, \boldsymbol{R}_T} \| P_{\mathbf{W}} \otimes P_{\boldsymbol{R}_T})$ is the mutual information and $D_{KL}$ is the KL divergence.*

This generalizes the information-theoretic generalization bound from i.i.d. settings Xu & Raginsky (2017) to time series data by assuming the residual sequence is stationary and $\beta$-mixing. We emphasize that this assumption is mild and widely used in the time series literature (Kreuzer et al., 2025; Dudek, 2022)—residuals are often approximately stationary after removing trends and seasonal components. Furthermore, $\beta$-mixing encompasses a broad class of dependent processes (e.g., ARMA, GARCH), making our result theoretically meaningful. While strict stationarity may not always hold, our method remains applicable and effective, even under mild violations (See Corollary 10). Formal definitions for assumptions and proof sketches are given in Appendix A.2. Following Yao et al. (2024), the mutual information term can be bounded in terms of LoRA's architecture:

**Corollary 3.** *With the same assumptions and integer $m$ in Theorem 2, the following inequality holds*

$$\mathbb{E}[L_{\mathcal{P}}(\mathbf{W}) - L_{\boldsymbol{R}_T}(\mathbf{W})] \leq \sqrt{6\sigma^2 m^{-1} qr \sum_{i \in \mathcal{I}} (d_{in} + d_{out})}, \qquad (6)$$

*where LoRA is applied to $\mathbf{W}_q^i, \mathbf{W}_k^i, \mathbf{W}_v^i \in \mathbb{R}^{d_{in} \times d_{out}}$, with total rank $r$ and quantization level $q$ bits.*

This bound demonstrates that the expected generalization error is controlled by the number of fine-tuned parameters, which is substantially smaller in LoRA compared to full-model tuning. By imposing a low-rank structure, LoRA effectively reduces the model's degrees of freedom, thereby implicitly lowering the mutual information between the training data and the learned parameters. From an information-theoretic perspective, reduced mutual information corresponds to improved stability and less overfitting, as the model becomes less sensitive to individual training sequences. While stronger guarantees (e.g., high-probability bounds) can be obtained under additional assumptions, the subgaussian stability bound already offers a compelling theoretical explanation for the generalization benefits. Importantly, this analysis can be naturally extended to other PEFT methods, since these techniques reduce mutual information by constraining the effective capacity of the model.

## 5 EXPERIMENTS

In this section, we evaluate the performance of Hopformer on a diverse set of time series forecasting benchmarks. Specifically, we address the following research questions: (i) Accuracy: How does Hopformer compare to state-of-the-art methods across a range of forecasting tasks? (ii) Ablation: What is the individual contribution of each key component of Hopformer to its overall performance? (iii) Robustness: How does Hopformer perform when varying context lengths and forecast horizons?

**Dataset and Baselines**: We evaluate Hopformer's performance on 6 datasets, including the Illness, EPF, and M5 benchmarks, along with three synthetic datasets (Sales1, Sales2, and Electricity). We restrict our evaluation to datasets with covariates, since our main contribution is to improve forecasting in covariate-driven settings; whereas in datasets without covariates Hopformer reduces to the backbone foundational models, and additional comparisons would not yield new insights. Dataset statistics are summarized in Table 1, where the total number of time steps is expressed as 'time steps per series × number of series'. The motivation for using the synthetic datasets (details are in Appendix SecB) is to illustrate the impact of covariates on time series forecasting while minimizing data leakage when comparing Hopformer with pre-trained foundational models for time series prediction. We benchmark Hopformer against Chronos-bolt, PatchTST, TemporalFusionTransformer, and two traditional statistical models, ARIMA and ETS. Please refer to the appendix for more details on the synthetic data.

Table 1: Statistics of Datasets used in Experiments.

| Dataset | Illness | EPF | M5 | Sale1 | Sale2 | Electricity |
|---|---|---|---|---|---|---|
| Covariates | 7 | 2 | 12 | 4 | 8 | 7 |
| Timesteps | 966 | 52 416 x 5 | 414 x 30 490 | 730 x 200 | 730 x 200 | 184 800 x 5 |

**Implementation and Evaluation Metrics:** We implement Hopformer using the AutoGluon library Shchur et al. (2023) and its usage is in Listing 1 (Appendix). We refer implementation details [1] to Appendix J. In the cross-sectional stage, the expert pool includes 8 regression models such as XG-Boost, LightGBM, Linear Regression, Random Forest, and CatBoost for processing future covariates, while a SimpleFeedForward network is used for past covariates. We use default hyperparameter for these regressors, and use ARIMA and ETS to capture lag and seasonal patterns, with max_ts_steps = 1000. We select 8 experts as a balanced pool: while adding more (up to 16) could increase diversity, the additional regressors showed weak performance under default settings and would likely require extensive hyperparameter tuning to be competitive. Due to time constraints, we did not pursue such tuning, and our experiments indicate that 8 well-performing experts already provide strong and efficient coverage.

For aggregation, we apply SPA (Equation 1) to Hopformer, and implement Equal Weighting, Single Best, and Linear Regression ($\hat{\omega}_{\boldsymbol{p}} \in \arg\min_{\pi \in \mathbb{R}^p} \|\mathbf{X} - \mathbf{Z}\pi\|_2^2$) for our ablation study. In the second

---

[1] Data and code are available at `https://www.dropbox.com/scl/fo/q4t08x79w1jxq2tnkz15d/ACXuC5nO6yS17cH696oaP0g?rlkey=e9ogypv287u8232l06dzn0u28&dl=0`.

stage, we use the Chronos-bolt-small (Chronos) to implement three models: a zero-shot model without fine-tuning, a fully fine-tuned model, and a model fine-tuned using LoRA. For LoRA, we employ a low-rank adaptation with rank = 8, a scaling factor of $\alpha = 16$, and a dropout rate of 5%, applied to the query, key, value, and output projections. We fine-tune models for 100 gradient steps. We run experiment on an Ubuntu server with $4 \times 1080$Ti GPUs and 96 CPUs.

We evaluate Hopformer using 20 rolling windows, each with a context length of 512 and a forecast horizon of 24, and report result in Table 2. We also report results under varying context lengths (32, 64, 128, 156, 512) with a fixed prediction length of 24, and varying prediction lengths (24, 71, 120) with a fixed context length of 256 in Tables 4 and 5. Specifically, we partition the dataset so that each test set consists of 20 consecutive prediction periods, with one prediction period reserved for validation, and the remaining data used for training. The latest 1,000 time steps serve to train the regressors and compute the SPA weights. Model performance is assessed by computing the Mean Absolute Scaled Error (MASE) and Mean Absolute Percentage Error (MAPE), averaged over all rolling windows. See Table 2 for our results. Note that MASE values in illness row are divided by 10 for readability.

Table 2: Forecasting performance of Hopformer, Cross-sectional regression module, Chornos-bolt-small (Chronos), and PatchTST (PTST), TemporalFusionTranformer (TFT), ARIMA, and ETS. Each cell reports MASE, and MAPE (lower is better). The bold numbers indicate the two lowest metric values in each row.

| Models | Hopformer | | | Cross-Sectional | | | | Chronos | | | DL and Stats | | | |
|---|---|---|---|---|---|---|---|---|---|---|---|---|---|---|
| Variants | 0-shot | Full | LoRA | SPA | Lasso | Best | Equal | 0-shot | Full | LoRA | PTST | TFT | Arima | Ets |
| Sale1 MASE | 0.946 | **0.761** | **0.819** | 1.592 | 1.598 | 1.646 | 1.610 | 1.095 | 0.927 | 0.971 | 0.911 | 0.883 | 1.191 | 1.136 |
| Sale1 MAPE | 0.915 | **0.631** | 0.686 | 1.482 | 1.483 | 1.510 | 1.523 | 0.951 | **0.679** | 0.707 | 0.881 | 0.813 | 1.029 | 1.380 |
| Sale2 MASE | 0.340 | **0.270** | **0.264** | 0.538 | 0.539 | 0.539 | 0.561 | 0.542 | 0.307 | 0.301 | 0.447 | 0.428 | 0.552 | 0.849 |
| Sale2 MAPE | 0.423 | **0.306** | **0.301** | 0.729 | 0.740 | 0.736 | 0.839 | 0.518 | 0.315 | 0.309 | 0.486 | 0.485 | 635 | 0.912 |
| Elec. MASE | 0.765 | **0.737** | **0.730** | 2.612 | 2.648 | 2.633 | 2.652 | 0.817 | 0.770 | 0.756 | 1.205 | 1.391 | 0.940 | 1.449 |
| Elec. MAPE | 0.078 | **0.075** | **0.075** | 0.263 | 0.266 | 0.257 | 0.259 | 0.083 | 0.079 | 0.078 | 0.143 | 0.164 | 0.106 | 0.171 |
| Illness MASE | 0.381 | **0.330** | **0.329** | 0.369 | 0.379 | 0.494 | 0.454 | 0.357 | 0.398 | 0.399 | 0.423 | 0.494 | 0.515 | 0.509 |
| Illness MAPE | 0.133 | **0.115** | **0.115** | 0.125 | 0.130 | 0.167 | 0.159 | 0.123 | 0.135 | 0.134 | 0.146 | 0.167 | 0.190 | 0.188 |
| EPF MASE | 0.654 | **0.650** | **0.642** | 1.279 | 1.760 | 0.813 | 17.17 | 0.662 | 0.674 | 0.720 | 1.466 | 0.862 | 0.895 | 1.091 |
| EPF MAPE | 1.114 | **1.112** | **1.109** | 2.554 | 3.027 | 1.804 | 11.67 | 1.180 | 1.252 | 1.265 | 3.252 | 1.689 | 1.383 | 1.124 |
| M5 MASE | 1.006 | **0.984** | 0.989 | 1.189 | 1.188 | 2.189 | 2.194 | 1.006 | 0.987 | 0.987 | 1.022 | **0.980** | 1.185 | 1.238 |
| M5 MAPE | 0.703 | 0.670 | **0.589** | 0.591 | 0.596 | 1.438 | 1.883 | 0.704 | 0.683 | 0.679 | 0.667 | 0.671 | 0.594 | **0.592** |

**Overall Performance**: Table 2 summarizes the performance of the forecasting models across six datasets. We highlight three key findings: (i) The best of the Hopformer variants outperforms baseline models, with an average relative improvement of 6.56% in MASE across all datasets. Performance gains are particularly notable on synthetic datasets with future covariates, achieving a 4.45% improvement in MAPE compared to the best baseline. (ii) In the zero-shot scenario, Hopformer attains comparable or superior results to Chronos in five out of six datasets, illustrating the efficacy of the SPA. Specifically, SPA achieves a 8.73% average reduction in MASE, highlighting its general compatibility with foundational models (visualized in Figure 2). (iii) Fine-tuning the residual module with LoRA achieves nearly identical performance to full fine-tuning (within 1% difference in MAPE), underscoring LoRA's efficiency by updating only targeted attention parameters.

**Ablation Study1**: We isolate and quantify the effect of Hopformer's aggregation strategies through an ablation study. We report the results in the cross-sectional columns of Tables 2, 4, and 5, where SPA consistently achieves the lowest MASE and MAPE across all datasets, reducing MASE by an average of 7.08% and MAPE by 6.99% compared to the next-best method. We further stress-test the strategies by varying the expert-pool size from 4 to 20 regressors (Figure 9), where SPA's advantage widens as the pool grows.

**Ablation Study2**: Table 3 investigates the effectiveness and generality of Hopformer's two-stage forecasting framework, particularly assessing the gain from its cross-sectional stage across time series foundation models—Chronos-bolt-small(Chronos), Moirai-small, Moirai-MoE-small, and Lag-Llama.

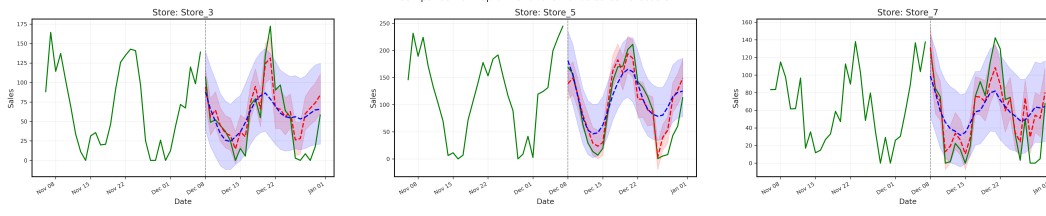

Figure 2: Zero-shot forecast comparison between Hopformer and Chronos on four representative stores from the SALES1 dataset. Blue lines show the ground-truth sales, while yellow and red lines depict model predictions.

Table 3: Forecasting performance (MASE and MAPE; lower is better) of four state-of-the-art foundation models—Chronos, Moirai, Moirai-MoE, and Lag-Llama—before and after integration with Hopformer's cross-sectional stage (denoted as "Hop"). "Uni" and 'Multi" denotes univariate and multivariate time series prediction. Experiments use 0-shot setting, a context length of 256, a forecast horizon of 24, and metric is averaged over 20 rolling windows. Moirais and Lag-Llama use 100 samples per forecast. Boldface highlights the best best-performing methods for each set of model variants. Highlighting Hop only for clearer view.

| Models | | Chronos | | Moirai | | | Moirai-moe | | | Lag-Llama | |
|---|---|---|---|---|---|---|---|---|---|---|---|
| Variants | | Uni | Hop | Uni | Mutli | Hop | Uni | Mutli | Hop | Uni | Hop |
| **EPF** | MASE | 0.785 | **0.732** | 0.952 | 0.915 | **0.821** | 0.886 | 0.845 | **0.768** | 1.168 | **0.773** |
| | MAPE | 1.414 | **1.355** | 1.245 | 1.427 | 1.550 | 1.363 | 1.312 | 1.580 | 2.387 | **1.626** |
| **Elec.** | MASE | 0.706 | **0.668** | 1.185 | 1.199 | **1.125** | 0.994 | 1.029 | **0.939** | 1.421 | **1.395** |
| | MAPE | 0.074 | **0.070** | 0.128 | 0.125 | **0.123** | 0.109 | 0.116 | **0.103** | 0.172 | **0.167** |
| **Sale1** | MASE | 0.577 | **0.576** | 1.274 | 1.187 | 1.337 | 2.721 | 1.923 | **1.878** | 1.217 | **1.183** |
| | MAPE | 0.562 | 0.562 | 1.217 | 1.225 | **1.073** | 0.738 | 0.798 | **0.714** | 1.013 | **0.885** |
| **Sale2** | MASE | 0.561 | **0.326** | 0.702 | 0.707 | **0.482** | 0.652 | 0.705 | **0.455** | 0.702 | **0.482** |
| | MAPE | 0.555 | **0.399** | 0.875 | 0.956 | **0.636** | 0.703 | 0.781 | **0.580** | 0.881 | **0.598** |

Three observations are particularly notable. **(i)** The inclusion of Hopformer's cross-sectional aggregation consistently improves forecasting performance across all four foundational models and datasets, highlighting its generalizability. For instance, on the EPF dataset, integrating the cross-sectional stage (Hop) reduces the Chronos MASE from 0.785 to 0.732 (5.3% improvement), Moirai from 0.952 (multi) to 0.821 (13.1%), Moirai-MoE from 0.886 (multi) to 0.768 (11.8%), and Lag-Llama from 1.168 to 0.773 (39.5%). **(ii)** These improvements underscore the robustness and broad compatibility of the Hopformer framework: regardless of differences in architectural complexity (e.g., MoE vs. standard transformer models), adding a dedicated cross-sectional aggregation stage effectively isolates and leverages covariate information. Such results demonstrate that the Hopformer paradigm—extracting and modeling residuals after explicit covariate adjustment—is a universally beneficial design choice, significantly boosting the predictive power of contemporary foundation models. **(iii)** Notably, Hopformer-enhanced models often outperform the native multivariate versions of Moirai and Moirai-MoE. For instance, on the Sale2 dataset, Hopformer applied to Chronos achieves a MASE of 0.326, compared to 0.707 and 0.705 from multivariate Moirai and Moirai-MoE respectively. This suggests that Hopformer offers a more effective and interpretable way to incorporate exogenous information than direct multivariate modeling. We visualize the forecasting results in SectionH.

**Robustness Analysis1**: Figure 3 probes Hopformer's robustness on zero-shot setting by sweeping (top) the context window and (bottom) the forecast horizon on three synthetic datasets. The result yields two main take-aways. (i) As the context window contracts, Hopformer retains most of its SPA-based advantage over Chronos. On Sales2, shrinking the context from 512 to 32 steps raises Chronos' MASE by 0.50, but Hopformer's by only 0.30. (ii) Both models deteriorate as the horizon grows, yet Hopformer remains ahead across all ranges because it starts from a lower error. On Sales2, the jump from 24 to 120-step forecasts adds 0.12 MASE to Hopformer and 0.05 to Chronos, but Hopformer's absolute error is still lower at every horizon. These patterns confirm that the SPA aggregation helps Hopformer preserve its advantage under both limited-context and long-range-forecast settings.

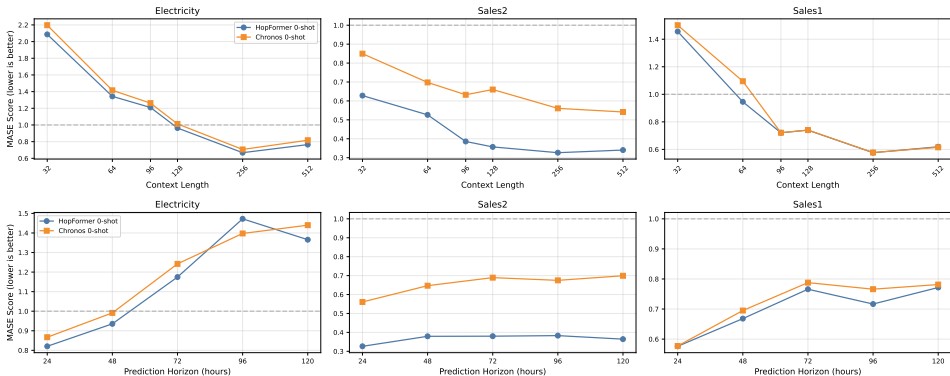

Figure 3: (Top) Effect of Context Length on Model Performance Across Datasets (prediction length = 24). (Bottom) Effect of Prediction Horizon on Model Performance Across Datasets (context length = 256).

**Robustness Analysis2**: We evaluate the robustness of Hopformer on the real-world EPF dataset (Wang et al., 2024) by varying the context length and prediction length. The results show that integrating Hopformer with Chronos consistently improves the performance of vanilla Chronos across all settings—zero-shot, full fine-tuning, and LoRA fine-tuning. Note that the performance of Hopformer variants differs slightly from Table 2, as we conducted additional hyperparameter tuning for the GBDT-based covariate regressors (e.g., XGBoost, LightGBM) to ensure a fairer comparison.

Table 4: Forecasting performance of Hopformer, Chronos-bolt-small (Chronos), and their hybrid variants on the EPF dataset across varying **context length**. Each cell reports MASE. Bold values highlight the two best-performing models per row. The prediction length is fixed at 24 and number of rolling window is 20. The number of gradient steps for full fine tuning and LoRA fine tuning is 100.

| **Models** | | **Hopformer** | | | **Cross-Sectional** | | | | **Chronos** | | |
|---|---|---|---|---|---|---|---|---|---|---|---|
| **Variants** | | 0-shot | Full | LoRA | SPA | Lasso | Best | Equal | 0-shot | Full | LoRA |
| **EPF** | 32 | 1.433 | **1.423** | **1.429** | 2.033 | 2.017 | 2.203 | 2.133 | 1.804 | 1.804 | 1.804 |
| | 64 | 0.908 | **0.901** | **0.904** | 1.343 | 1.327 | 1.513 | 1.444 | 1.100 | 1.091 | 1.096 |
| | 128 | 0.802 | **0.794** | **0.795** | 1.175 | 1.161 | 1.323 | 1.287 | 0.918 | 0.914 | 0.915 |
| | 256 | 0.732 | **0.725** | **0.727** | 1.135 | 1.116 | 1.218 | 1.174 | 0.785 | 0.787 | 0.788 |
| | 512 | 0.662 | **0.655** | **0.658** | 1.141 | 1.118 | 1.033 | 0.986 | 0.662 | 0.672 | 0.671 |

Table 5: Forecasting performance of Hopformer, Chronos-bolt-small (Chronos), and their hybrid variants on the EPF dataset across varying **prediction length**. Each cell reports MASE. Bold values highlight the two best values per row. The context length is fixed at 256. The number of rolling windows is 10 for prediction length of 24 and 72, and is 4 for 120. The number of gradient steps for full fine tuning and LoRA fine tuning is 100.

| **Models** | | **Hopformer** | | | **Cross-Sectional** | | | | **Chronos** | | |
|---|---|---|---|---|---|---|---|---|---|---|---|
| **Variants** | | 0-shot | Full | LoRA | SPA | Lasso | Best | Equal | 0-shot | Full | LoRA |
| **EPF** | 24 | 0.796 | **0.786** | **0.789** | 1.209 | 1.188 | 1.299 | 1.194 | 0.896 | 0.904 | 0.904 |
| | 72 | 0.875 | **0.864** | **0.866** | 1.625 | 1.601 | 1.317 | 1.293 | 1.043 | 1.022 | 1.036 |
| | 120 | **0.896** | 0.919 | **0.916** | 1.856 | 1.822 | 1.354 | 1.363 | 1.154 | 1.165 | 1.151 |

Table 4 summarizes the Mean Absolute Scaled Error (MASE) of Hopformer (built on top of Chronos) and vanilla Chronos when the available context ranges from 32 to 512 time steps. Table 5 summarizes the MASE of the models across varying prediction length. Four findings stand out. (i)**Zero-shot performance**: Hopformer consistently beats Chronos at every context length, with the largest gain ($\sim 37.1\%$) at the shortest window of 32 steps, as visualized in Figure 8. This suggests that the covariate-driven expert pool provides valuable signal when historical information is scarce. (ii) **Fine-tuned performance**: After full-parameter or LoRA fine-tuning, Hopformer still yields lower error than Chronos. Removing covariate effects in the first stage appears to simplify the residual dynamics, making the subsequent transformer easier to adapt. (iii) **Cross-sectional aggregation**: In this two-covariate setting, SPA and Lasso deliver comparable accuracy, indicating that with a very small covariate set the sparsity prior in SPA offers little advantage over a standard $\ell_1$-penalised regression. (iv) **Long-horizon prediction**: Hopformer also surpasses Chronos at every horizon, as

visualized in Figure 8 (Appendix). The largest gain ($\sim 25.8\%$) locates at the longest horizon of 120 steps, again demonstrating the value of the covariate-driven expert pool as the prediction window expands.

**Visualization of Forecasting**: Figure 4 illustrates Hopformer's prediction process on the Sales1 dataset, providing an intuitive example of how the model leverages covariates during forecasting. To complement this, Figures 10 and 11 visualize the ablation results from Table 3, highlighting Hopformer's advantages on both the Sales1 and EPF datasets, where covariates play a key role in improving prediction accuracy.

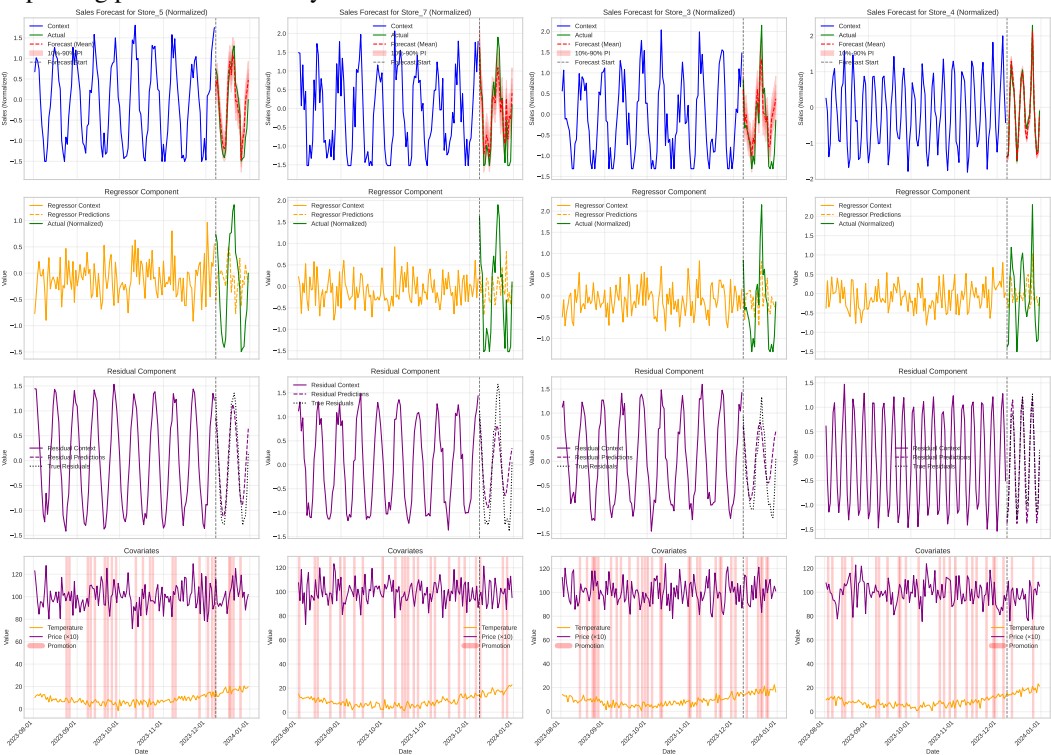

Figure 4: Decomposition of Hopformer's prediction pipeline on the SALE1 dataset. Each column corresponds to a store. **Row 1**: ground truth and final forecast; **Row 2**: cross-sectional stage (covariate aggregation); **Row 3**: residual transformer output; **Row 4**: covariate trajectories. Removing covariate effects (Row 1 → 3) yields a smoother, quasi-periodic residual series, illustrating the division of labour between the two stages.

## 6 DISCUSSION AND CONCLUSION

We proposed Hopformer, a novel two-stage forecasting framework designed for time series with high-dimensional covariates. The first stage performs trend extraction via sparse pattern aggregation, which adaptively combines a pool of regressors. This stage operates as a homogeneity pursuit mechanism, projecting multiple time series into a residual space with reduced structural variation. Theoretically, we establish an oracle inequality showing that SPA achieves near-optimal prediction relative to the best sparse expert combination, with a provable complexity-penalized bound. The second stage learns residual dependencies via Transformer fine-tuning. By using LoRA, we minimize the number of trainable parameters while retaining model flexibility. Under mild assumptions on temporal dependence and loss stability, we derive a generalization bound based on information-theoretic complexity, showing that PEFT methods such as LoRA reduce the mutual information between training data and model weights. Empirically, these two components enable Hopformer to deliver accurate, robust, and efficient forecasting across a wide range of time series tasks. Several directions remain open for exploration. First, exploring alternative parameter-efficient fine-tuning approaches beyond LoRA both theoretically and empirically could enhance flexibility in various scenarios. Second, incorporating task-aware pretraining strategies into the residual modeling may further improve cross-task generalization.

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

# A APPENDIX

## A.1 PROOF OF THEOREM 1

**Lemma 4** ((Leung & Barron, 2006)). *Let $M$ be a finite set of candidate models with $|M|$ denoting its cardinality. For each model $m \in M$, define $\hat{r}_m = \mathbb{E} \|\hat{g}_m - \eta\|^2$ as the empirical risk estimate of model $m$.*

*Define the aggregation weights $w_m$ are given by:*

$$w_m = \frac{\pi_m \exp(-\hat{r}_m/4)}{\sum_{m' \in M} \pi_{m'} \exp(-\hat{r}_{m'}/4)}, \tag{7}$$

*where $\pi_m = \exp(-C_m)$ satisfying $\sum_{m \in M} \pi_m \leq 1$ with $C_m$ denote complexity penalty term associated with model $m$.*

*Then, the unbiased risk estimate for $\hat{\mu}$ satisfies:*

$$\hat{r} = \sum_{m \in M} w_m \hat{r}_m < \min_{m \in M} \{\hat{r}_m + 4C_m\} \tag{8}$$

*Proof.* Let $\hat{m} \in \arg\min_{m \in M} \{\hat{r}_m + 4C_m\}$. Then, we can write:

$$\hat{r}_m = 4 \left[ \log \frac{\pi_m}{w_m} - \log \sum_{m'} \pi_{m'} \exp(-\hat{r}_{m'}/4) \right] = \hat{r}_{\hat{m}} + 4 \left[ C_{\hat{m}} - \log \frac{w_m}{\pi_m} + \log w_{\hat{m}} \right].$$

Therefore:

$$\hat{r} = \sum_{m \in \mathcal{M}} w_m \hat{r}_m = \hat{r}_{\hat{m}} + 4 \left[ C_{\hat{m}} - D(w\|\pi) + \log w_{\hat{m}} \right] < \min_{m \in \mathcal{M}} \{\hat{r}_m + 4C_m\}.$$

$\square$

The proof of Theorem 1 follows from letting

$$\hat{r}_{\boldsymbol{p}} = \frac{1}{\sigma^2} \sum_{k=1}^{n} \left( X_{t,k}^{(i)} - g_{\hat{\omega}_{\boldsymbol{p}}}(\mathbf{Z}_{t,k}^{(i)}) \right)^2 + 2|\mathbf{p}|$$

## A.2 PROOF OF THEOREM 2

**Lemma 5** (Xu & Raginsky (2017)). *Consider a pair of random variables $X$ and $Y$ with joint distribution $P_{X,Y}$. Let $\bar{X}$ be an independent copy of $X$, $\bar{Y}$ be an independent copy of $Y$, such that $P_{\bar{X},\bar{Y}} = P_X \otimes P_Y$. Let $f$ be $\sigma$-subgaussian under $P_{\bar{X},\bar{Y}} = P_X \otimes P_Y$, then*

$$\left| \mathbb{E}[f(X,Y)] - \mathbb{E}[f(\bar{X},\bar{Y})] \right| \leq \sqrt{2\sigma^2 I(X;Y)},$$

Let $X = \boldsymbol{R}_T$, $Y = W$ and $f(s,w) = \frac{1}{T-d} \sum_{i=d+1}^{T} \ell(w,x_i)$, then we can express the empirical risk as $L_{\boldsymbol{R}_T}(w) = f(\boldsymbol{R}_T, w)$, and the population risk $L_{\mathcal{P}}(w) = \mathbb{E}_{\mathcal{P}}[f(\boldsymbol{R}_T, w)]$.

Thus, the generalization error reformulates as

$$\mathbb{E}[L_{\mathcal{P}}(\mathbf{W}) - L_{\boldsymbol{R}_T}(\mathbf{W})] = \mathbb{E}[f(\bar{\boldsymbol{R}}_T, \bar{\mathbf{W}})] - \mathbb{E}[f(\boldsymbol{R}_T, \mathbf{W})].$$

**Definition 6** (Stationarity). *A random sequence $S_\infty$ is stationary when all its finite-dimensional distributions are time-invariant: for all $t$ and all non-negative integers $i$ and $j$, the random vectors $S_{t:t+i}$ and $S_{t+j:t+i+j}$ have the same distribution.*

**Definition 7** ($\beta$-Mixing). *Consider a stationary random sequence $S_\infty$ defined on a probability space $(\Omega, \Sigma, P_\infty)$. Denote $S_{i:j} := (S_i, S_{i+1}, \ldots, S_j)$, $S_\infty := S_{-\infty:\infty}$ an infinite dimensional sequence. Denote $\mathbb{P}_{i:j}$ and $\mathbb{P}_\infty$ as the associated joint distributions, and $\sigma_{i:j} = \sigma(S_{i:j})$ and $\sigma_\infty = \sigma(S_\infty)$ as the $\sigma$-fields. Let $P_0$ be the restriction of $P_\infty$ to $\sigma_{-\infty:0}$, $P_a$ be the restriction of $P_\infty$ to $\sigma_{a:\infty}$, and $P_{0\otimes a}$ be the restriction of $P_\infty$ to $\sigma(S_{\infty:0}, S_{a:\infty})$. The coefficient of absolute regularity, or $\beta$-mixing coefficient, $\beta_a$, is given by*

$$\beta_a := \|P_0 \times P_a - P_{0\otimes a}\|_{TV},$$

*where $\|\cdot\|_{TV}$ is the total variation norm. A stochastic process is absolutely regular, or $\beta$-mixing, if $\beta_a \to 0$ as $a \to \infty$.*

**Lemma 8.** *Assuming the loss function $\ell(w, R)$ is $\sigma$-subgaussian, if the target time series is stationary and $\beta$-mixing, then there exists a constant $a > 0$ and an integer $m$ such that $2am \leq T$ ensuring that the empirical loss $f(\boldsymbol{R}_T, w)$ is $\frac{\sigma}{\sqrt{m}}$-subgaussian.*

*Proof.* We divide the sequence $S_{d+1:T}$ into $2m$ blocks of length $a$, such that $2ma + d = T$. Identify the blocks as follows:

$$U_j = \{R_i : 2(j-1)a + d + 1 \leq i \leq (2j-1)a + d\},$$
$$V_j = \{R_i : (2j-1)a + d + 1 \leq i \leq 2ja + d\}.$$

Let $\mathbf{U}$ be the sequence of odd blocks, and let $\mathbf{V}$ be the sequence of even blocks. With the mixing conditions (need $\beta$-mixing assumption), we choose $a$ large enough so that the odd blocks are almost independent, but at the same time small enough so that the odd blocks behave similarly to the original mixing sequence. Let $\mathbf{U}'$ be a sequence of identically distributed independent blocks such that each block has the same distribution as a block from the original sequence:

$$\mathcal{L}(U_j') = \mathcal{L}(U_j) = \mathcal{L}(U_1).$$

By the linearity of subgaussianity, the block-wise empirical average is given by:

$$\hat{f}_{U_j}(w) = \frac{1}{a} \sum_{i \in U_j} \ell(w, R_i)$$

remains $\sigma$-subgaussian. By Yu (1994), for the odd blocks sequence $f_{\mathbf{U}}(w) = \frac{1}{m} \sum_{j=1}^{m} \hat{f}_{U_j}(w)$, we have

$$\log \mathbb{E} \left[ e^{\lambda(f_{\mathbf{U}}(w) - \mathbb{E}(f_{\mathbf{U}}(w)))} \right] \lesssim \log \mathbb{E} \left[ e^{\lambda(f_{\mathbf{U}'}(w) - \mathbb{E}(f_{\mathbf{U}'}(w)))} \right] \leq \frac{\lambda^2 \sigma^2}{2m},$$

which implies that $f_{\mathbf{U}}(w)$ is $\frac{\sigma}{\sqrt{m}}$-subgaussian.

Applying the same construction to the sequence of even blocks, the total empirical loss $f(\boldsymbol{R}_T, w)$ can be viewed as the sum of two $\frac{\sigma}{\sqrt{m}}$-subgaussian variables. By the linearity property of subgaussianity, it follows that $f(\boldsymbol{R}_T, w)$ is also $\frac{\sigma}{\sqrt{m}}$-sub-Gaussian.

$\square$

**Remark 9.** *While Lemma 8 assumes strict stationarity and $\beta$-mixing to control the concentration of empirical risk, similar results can be extended to locally stationary time series with bounded local mixing coefficients. In such cases, the subgaussian property holds approximately within short windows, potentially with additional approximation error terms.*

Applying Lemma 5 and Lemma 8, we obtain the generalization bound as in equation 5

$$\mathbb{E}[L_{\mathcal{P}}(\mathbf{W}) - L_{\boldsymbol{R}_T}(\mathbf{W})] \leq \sqrt{\frac{2\sigma^2}{m} I(\boldsymbol{R}_T; \mathbf{W})}.$$

We follow the blocking technique from the proof above to divide $S_{d+1:N}$ into $2m$ blocks of length $a$, ensuring that sufficiently spaced blocks are approximately independent under the assumption of mixing.

**Corollary 10** (Generalization Bound Under Local Stationarity). *Let the residual sequence $\{R_t\}_{t=1}^{T}$ be **locally stationary** in the sense of Dahlhaus (1997): there exists a family of stationary processes $\{\boldsymbol{R}_t^{(u)}\}_{u \in [0,1]}$ such that*

$$\sup_t \mathbb{E} \left| \boldsymbol{R}_t - \boldsymbol{R}_t^{(t/T)} \right| \leq \delta_T,$$

*where $\delta_T \to 0$. Assume each $\{\boldsymbol{R}_t^{(u)}\}$ is $\beta$-mixing with mixing coefficients uniformly bounded by $\beta(m)$.*

*Then there exist constants $a > 0$ and $m$ with $2am \leq T$ such that*

$$\mathbb{E}\left[L_{\mathcal{P}}(\mathbf{W}) - L_{\boldsymbol{R}_T}(\mathbf{W})\right] \leq \sqrt{2\sigma^2 m^{-1} I(\boldsymbol{R}_T; \mathbf{W})} + C\delta_T,$$

*for a constant $C$ independent of $T$.*

By the data processing inequality, we have

$$I(\mathbf{W} + \Delta\mathbf{W}; \boldsymbol{R}_T \mid P_{\mathbf{W}|\boldsymbol{R}_T}) \leq I(\Delta\mathbf{W}; \boldsymbol{R}_T \mid P_{\mathbf{W}|\boldsymbol{R}_T}, \mathbf{W}).$$

Using the chain rule and the assumption that $\mathbf{W}$ is independent of $\boldsymbol{R}_T$, the mutual information can be decomposed as

$$I(\mathbf{W}; \boldsymbol{R}_T \mid P_{\mathbf{W}|\boldsymbol{R}_T}) + I(\Delta\mathbf{W}; \boldsymbol{R}_T \mid P_{\mathbf{W}|\boldsymbol{R}_T}, \mathbf{W}) = I(\Delta\mathbf{W}; \boldsymbol{R}_T \mid P_{\mathbf{W}|\boldsymbol{R}_T}, \mathbf{W}).$$

Combining these results with 5, and apply the standard inequality that mutual information is always upper bounded by entropy ($I\left(\left\{\mathbf{W}_q^i, \mathbf{W}_k^i, \mathbf{W}_v^i\right\}_{i\in\mathcal{I}}; \boldsymbol{R}_T\right) \leq H\left(\left\{\mathbf{W}_q^i, \mathbf{W}_k^i, \mathbf{W}_v^i\right\}_{i\in\mathcal{I}}\right)$), we obtain the generalization bound

$$
\begin{aligned}
\mathbb{E}[L_{\mathcal{P}}(\mathbf{W}) - L_{\boldsymbol{R}_T}(\mathbf{W})] &\leq \sqrt{\frac{2\sigma^2}{m} I(\Delta\mathbf{W}; \boldsymbol{R}_T \mid P_{\mathbf{W}|\boldsymbol{R}_T}, \mathbf{W})} \\
&= \sqrt{\frac{2\sigma^2}{m} I\left(\left\{\mathbf{W}_q^i, \mathbf{W}_k^i, \mathbf{W}_v^i\right\}_{i\in\mathcal{I}}; \boldsymbol{R}_T \mid P_{\mathbf{W}_{\{q,k,v\}}|\boldsymbol{R}_T}, \mathbf{W}\right)} \\
&\leq \sqrt{\frac{2\sigma^2}{m} H\left(\left\{\mathbf{W}_q^i, \mathbf{W}_k^i, \mathbf{W}_v^i\right\}_{i\in\mathcal{I}}\right)} \\
&\leq \sqrt{\frac{6\sigma^2}{m} qr \sum_{i\in\mathcal{I}} (d_{in} + d_{out})},
\end{aligned}
$$

where $\mathbf{W}_q^i, \mathbf{W}_k^i, \mathbf{W}_v^i \in \mathbb{R}^{d_{in}\times d_{out}}$. The last inequality holds because entropy is upper bounded by the uniform distribution over its possible support set.

**Remark.** For a more general setting that each attention head $i \in \mathcal{I}$ with:

- Query matrix: $\mathbf{W}_q^i \in \mathbb{R}^{d_{\text{in}}\times d_k}$,
- Key matrix: $\mathbf{W}_k^i \in \mathbb{R}^{d_{\text{in}}\times d_k}$,
- Value matrix: $\mathbf{W}_v^i \in \mathbb{R}^{d_{\text{in}}\times d_v}$.

Thus, the total entropy bound is:

$$H\left(\left\{\mathbf{W}_q^i, \mathbf{W}_k^i, \mathbf{W}_v^i\right\}_{i\in\mathcal{I}}\right) \leq qr \sum_{i\in\mathcal{I}} \left[2(d_{\text{in}} + d_k) + (d_{\text{in}} + d_v)\right].$$

Applying the mutual information bound, we conclude:

$$I\left(\left\{\mathbf{W}_q^i, \mathbf{W}_k^i, \mathbf{W}_v^i\right\}_{i\in\mathcal{I}}; \boldsymbol{R}_T\right) \leq qr \sum_{i\in\mathcal{I}} \left[2(d_{\text{in}} + d_k) + (d_{\text{in}} + d_v)\right].$$

**Remark.** Consider a pre-trained model with weights $\mathbf{W}_0$ (independent of $\boldsymbol{R}_T$) and LoRA's learned low-rank parameters $\Delta\mathbf{W} = \mathbf{BA}$. Since $\mathbf{W} = \mathbf{W}_0 + \Delta\mathbf{W}$ and $\mathbf{W}_0$ are fixed, $I(\boldsymbol{R}_T; \mathbf{W}) = I(\boldsymbol{R}_T; \Delta\mathbf{W})$. If $\Delta\mathbf{W}$ contains $d$ parameters in total (with a suitably discretized or bounded range), an upper bound on its entropy is $H(\Delta\mathbf{W}) \approx d \log |\text{Range}|$. For low-rank $\mathbf{A} \in \mathbb{R}^{m\times r}, \mathbf{B} \in \mathbb{R}^{r\times n}$, $d = r(m + n)$ which is much smaller than the full model's parameter count $mn$. Thus one can derive $I(\boldsymbol{R}_T; \mathbf{W}) \leq H(\Delta\mathbf{W}) = \mathcal{O}(r(m + n))$ (in bits). As $r, m, n$ are modest for LoRA, the information bound is dramatically smaller. This back-of-the-envelope derivation aligns with the idea that LoRA's limited parameter count yields a provably smaller $I(\boldsymbol{R}_T; \mathbf{W})$, reinforcing why it generalizes well under the stability condition.

## B  SYNTHETIC DATA

SYNTHETIC SALE DATA - SALE1

We generate synthetic store sales data across 200 stores, each parameterized by independent and identically distributed (i.i.d.) parameters drawn from normal distributions. Each store $i$ has unique

parameters including amplitude $A_i$, frequency $f_i$, baseline sales $B_i$, and covariate sensitivities such as promotion effect $P_i$, temperature effect $T_i$, price effect $C_i$, promotion probability $p_i$, temperature noise $\sigma_{T_i}$, and sales noise $\sigma_{S_i}$. The mean and standard deviation of each of the parameters are in Table 6. The sales data for each store at time $t$ are generated as follows:

$$\text{Sales}_{i,t} = B_i + A_i \cdot \sin\left(\frac{2\pi t}{f_i}\right) + P_i \cdot \text{promotion}_{i,t} + T_i \cdot \text{temperature}_{i,t} + C_i \cdot \text{price}_{i,t} + \epsilon_{i,t}, \quad \epsilon_{i,t} \sim \mathcal{N}(0, \sigma_{S_i}^2)$$

Here, the promotion occurrence is sampled using the store-specific promotion probability $p_i$. The temperature follows a seasonal pattern with added Gaussian noise parameterized by $\sigma_{T_i}$. The term $\varepsilon_{i,t}$ represents Gaussian noise with standard deviation $\sigma_{S_i}$, capturing real-world randomness in sales. The covariates (promotion, temperature, and price) are known in advance and used as inputs for forecasting.

Table 6: Mean and standard deviation of store-specific parameters

|  | $A$ | $f$ | $B$ | $P$ | $T$ | $C$ | $p$ | $\sigma_T, \sigma_S$ |
|---|---|---|---|---|---|---|---|---|
| Mean | 80 | {7,14,30,90} | 200 | 30 | 0.3 | -12 | 0.20 | 2.0, 10.0 |
| Std. Dev. | 30 | – | 80 | 10 | 0.1 | 5 | 0.05 | 0.5, 3.0 |

Figure 5 illustrates key aspects of the synthetic SALE1 dataset. The left panel shows the distribution of store-specific parameters such as promotion effects, baseline sales, and trend coefficients, highlighting the diversity across stores. The right panel presents sample sales trajectories from randomly selected stores, demonstrating realistic seasonality and promotion-driven variability embedded in the generated data.

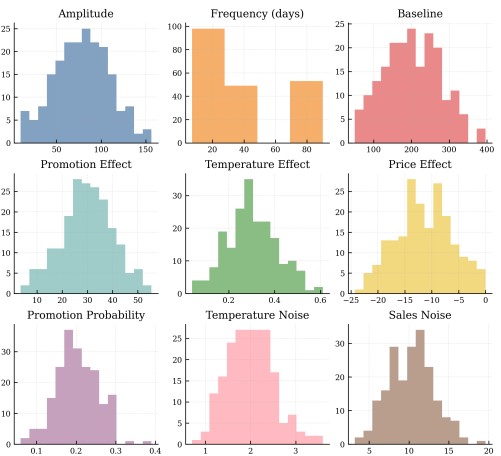

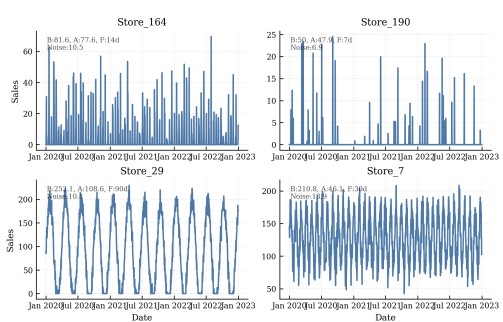

(a) Distribution of store-specific parameters used for synthetic data (SALE1) generation (e.g., promotion effect, baseline sales, trend).

(b) Sample sales trajectories from randomly selected stores in SALE1. Seasonal and promotional effects are clearly visible.

Figure 5: Illustration of synthetic dataset (SALE1) generation. Left: parameter distribution across stores. Right: sample time series showing temporal dynamics.

### SYNTHETIC SALE DATA - SALE2

We simulate realistic sales patterns across 200 stores by parameterizing each store with a high-dimensional vector of behavioral and structural attributes sampled independently. Each is parameterized by independent and identically distributed (i.i.d) parameters drawn from normal distributions, where the mean and standard deviation are shown in Table 7. For each store $i$, we define the daily

sales $y_{i,t}$ as a combination of trends, seasonality, covariate interactions, and stochastic noise:

$$y_{i,t} = \underbrace{B_i + \alpha_i t + A_i \sin\left(\frac{2\pi t}{f_i}\right) + S_i \sin\left(\frac{2\pi t}{365}\right)}_{\text{trend + intra-cycle + annual seasonality}} + E_{i,t}^{\text{cov}} + \varepsilon_{i,t}$$

Here, $B_i$ is the baseline sales, $\alpha_i$ is the trend coefficient, $A_i$ is the amplitude of intra-cycle seasonality with frequency $f_i$, and $S_i$ controls the strength of annual seasonality. The term $\varepsilon_{i,t} \sim \mathcal{N}(0, \sigma_{S_i}^2)$ is Gaussian noise. The covariate effect $E_{i,t}^{\text{cov}}$ incorporates multiple nonlinear and interactive effects:

$$E_{i,t}^{\text{cov}} = \beta_{\text{promo}} \cdot \text{promo}_{i,t} + \beta_{\text{holiday}} \cdot \text{holiday}_t + \beta_{\text{weekend}} \cdot \text{weekend}_t + \beta_{\text{competitor}} \cdot \text{comp\_promo}_{i,t}$$
$$+ \beta_{\text{inventory}} \cdot \text{inventory}_{i,t} + m_{i,t} + \beta_{\text{temp}} \cdot (1 - |T_{i,t} - T_i^*|) + \beta_{\text{price}} \cdot \text{price}_{i,t}^{\eta_i}$$

where: $\text{promo}_{i,t}$, $\text{holiday}_t$, and $\text{weekend}_t$ are binary indicators for promotion, holiday, and weekend; $\text{comp\_promo}_{i,t}$ indicates competitor promotions; $\text{inventory}_{i,t}$ and $m_{i,t}$ represent inventory effects and exponentially decayed marketing memory, respectively; $T_{i,t}$ is the daily temperature and $T_i^*$ is the store-specific optimal temperature; $\text{price}_{i,t}^{\eta_i}$ models nonlinear price elasticity with elasticity coefficient $\eta_i$. These represents known covariates in the dataset.

Table 7: Parameter distributions used for complex synthetic data generation.

| Symbol | $\mathcal{A}$ | $\mathcal{B}$ | $f$ | $\beta_{\text{promo}}$ | $\beta_{\text{weekend}}$ | $\beta_{\text{holiday}}$ | $\beta_{\text{temp}}$ | $T^*$ | $\beta_{\text{price}}$ | $\eta$ |
|---|---|---|---|---|---|---|---|---|---|---|
| Mean | 80 | 200 | {7,14,30,90} | 30 | 15 | 40 | 0.3 | 22 | -12 | 1.5 |
| Std | 30 | 80 | — | 10 | 5 | 15 | 0.1 | 3 | 5 | 0.3 |

| Symbol | $\beta_{\text{comp}}$ | $\beta_{\text{mkt}}$ | $\lambda$ | $\beta_{\text{inv}}$ | $S$ | $\alpha$ | $p$ | $p_{\text{comp}}$ | $\sigma_T$ | $\sigma_S$ |
|---|---|---|---|---|---|---|---|---|---|---|
| Mean | -15 | 0.4 | 0.85 | 0.2 | 30 | 0.01 | 0.2 | 0.15 | 2.0 | 10.0 |
| Std | 5 | 0.1 | 0.05 | 0.05 | 10 | 0.005 | 0.05 | 0.05 | 0.5 | 3.0 |

Table 6 illustrates key aspects of the synthetic SALE2 dataset. The left panel shows the distribution of store-specific parameters such as promotion effects, baseline sales, and trend coefficients, highlighting the diversity across stores. The right panel presents sample sales trajectories from randomly selected stores, demonstrating realistic seasonality and promotion-driven variability embedded in the generated data.

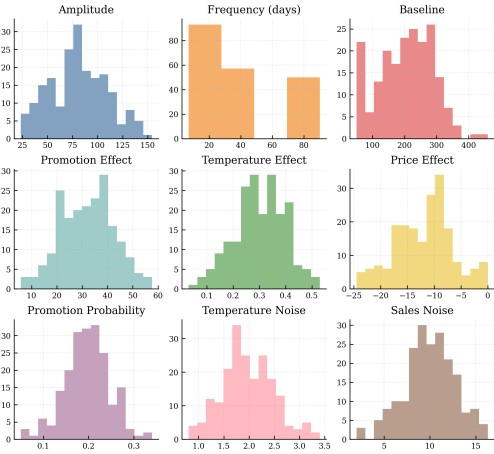
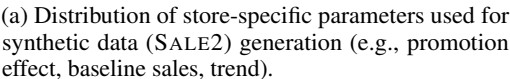
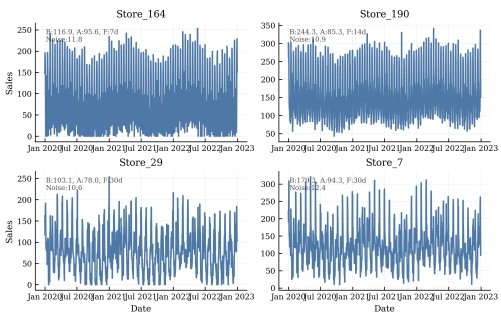

(a) Distribution of store-specific parameters used for synthetic data (SALE2) generation (e.g., promotion effect, baseline sales, trend).

(b) Sample sales trajectories from randomly selected stores in SALE2. Seasonal and promotional effects are clearly visible.

Figure 6: Illustration of synthetic dataset (SALE2) generation. Left: parameter distribution across stores. Right: sample time series showing temporal dynamics.

SYNTHETIC ELECTRICITY LOAD GENERATION

We simulate regional electricity load patterns using a high-fidelity parametric model that incorporates weather, calendar, infrastructure, and market dynamics. For each region $r$ and time $t$, the electricity load $L_{r,t}$ is modeled as:

$$L_{r,t} = \underbrace{B_r + \alpha_r t}_{\text{trend}} + \underbrace{D_r(t) + W_r(t) + Y_r(t)}_{\text{daily, weekly, yearly patterns}} + E_{r,t}^{\text{cov}} + \varepsilon_{r,t}$$

Here, $B_r$ is the base load, $\alpha_r$ is the regional trend (load growth), and $D_r(t)$, $W_r(t)$, $Y_r(t)$ are the daily, weekly, and yearly seasonal patterns. The covariate component $E_{r,t}^{\text{cov}}$ accounts for complex external influences:

$$E_{r,t}^{\text{cov}} = \beta_{\text{temp}} \cdot f_T(T_{r,t}) + \beta_{\text{humid}} \cdot H_{r,t} + \beta_{\text{wind}} \cdot W_{r,t} + \beta_{\text{solar}} \cdot S_{r,t}$$
$$+ \beta_{\text{weekend}} \cdot \text{Weekend}_t + \beta_{\text{holiday}} \cdot \text{Holiday}_t + \beta_{\text{DST}} \cdot \text{DST}_t + \beta_{\text{outage}} \cdot \text{PlannedOutage}_t$$

The function $f_T(\cdot)$ captures the nonlinear sensitivity to temperature using region-specific parameters (e.g., piecewise or quadratic effects). Additional noise $\varepsilon_{r,t} \sim \mathcal{N}(0, \sigma_r^2)$ accounts for random fluctuations. Additionally, the nonlinear effect of temperature on load is modeled using asymmetric thresholds for heating and cooling:

$$f_T(T) = \gamma_{\text{cool}} \cdot \max(0, T - T^C)^2 + \gamma_{\text{heat}} \cdot \max(0, T^H - T)^2,$$

where: $T$ is the observed temperature; $T^C$ is the cooling threshold (e.g., 22°C); $T^H$ is the heating threshold (e.g., 15°C); $\gamma_{\text{cool}}$, $\gamma_{\text{heat}}$ are the sensitivity coefficients for cooling and heating loads. This formulation captures the fact that electricity demand rises nonlinearly when temperatures deviate from the comfort band.

For market simulation, we also define a dynamic electricity price:

$$P_{r,t} = \beta_{\text{base}} + \beta_{\text{peak}} \cdot \mathbb{I}[L_{r,t} > \text{Cap}_r]^{\gamma_r} + \nu_{r,t}$$

Here, $\mathbb{I}[L_{r,t} > \text{Cap}_r]$ is an indicator for capacity exceedance, with exponent $\gamma_r$ modeling price spikes, and $\nu_{r,t} \sim \mathcal{N}(0, \sigma_P^2)$ denotes price noise. The framework supports renewable volatility and substitution through additional interaction terms.

Table 8 summarizes the parameter ranges used to generate realistic electricity load and price patterns across regions. These ranges govern variability in demand behavior, weather sensitivity, calendar effects, and market responses, enabling the simulation of heterogeneous grid conditions reflective of residential, commercial, industrial, and mixed-use regions.

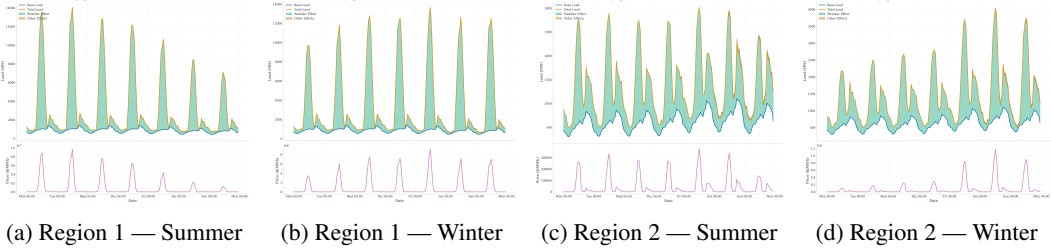

| (a) Region 1 — Summer | (b) Region 1 — Winter | (c) Region 2 — Summer | (d) Region 2 — Winter |

Figure 7: Load decomposition over one representative week for two regions across summer and winter. Each panel visualizes base demand, weather influence, calendar signals, and total load.

Figure 7 visualizes the weekly decomposition of electricity load across two regions under contrasting seasonal conditions. The top row shows Region1's load components during summer and winter weeks, while the bottom row depicts the same for Region2. These plots highlight the distinct roles of temperature, calendar effects, and renewable volatility in shaping regional demand, and emphasize how seasonal dynamics interact differently across geographic and structural profiles.

## C  ROBUSTNESS ON REAL-WORLD DATASETS

We evaluate the robustness of Hopformer on the real-world EPF dataset (Wang et al., 2024) by varying the context length and prediction length. The results show that integrating Hopformer with

Table 8: Parameter ranges used to generate region-specific electricity load and price patterns.

| Category | Symbol | Min | Max |
|---|---|---|---|
| **Base Load & Seasonality** | $B_r$ (base load, MW) | 500 | 5000 |
| | $A_{\text{year}}$ (yearly amplitude) | 0.15 | 0.35 |
| | $\alpha_r$ (load trend) | -0.02 | 0.04 |
| **Temperature Sensitivity** | $T^*$ (optimal temp) | 18.0 | 23.0 |
| | $T^C$ (cooling threshold) | 22.0 | 27.0 |
| | $T^H$ (heating threshold) | 12.0 | 18.0 |
| | $\gamma_{\text{cool}}$ (cooling slope) | 0.02 | 0.06 |
| | $\gamma_{\text{heat}}$ (heating slope) | 0.01 | 0.04 |
| **Weather Sensitivity** | $\beta_{\text{humid}}$ | 0.001 | 0.005 |
| | $\beta_{\text{wind}}$ | -0.005 | 0.002 |
| | $\beta_{\text{solar}}$ | -0.02 | -0.005 |
| **Calendar Effects** | $\beta_{\text{weekend}}$ | -0.25 | -0.05 |
| | $\beta_{\text{holiday}}$ | -0.30 | -0.10 |
| | $\beta_{\text{DST}}$ | -0.05 | 0.05 |
| **Infrastructure Effects** | $\beta_{\text{outage}}$ | -0.20 | -0.05 |
| **Renewables** | $\beta_{\text{renew}}$ (substitution) | 0.20 | 0.80 |
| | $V_{r,t}$ (volatility) | 0.01 | 0.05 |
| **Price Model** | $\beta_{\text{base}}$ (price base) | 20 | 60 |
| | $\beta_{\text{peak}}$ (peak multiplier) | 1.5 | 4.0 |
| | $\sigma_P$ (price volatility) | 0.05 | 0.20 |
| **Capacity Constraints** | $\text{Cap}_r$ | 0.80 | 0.95 |
| | $\gamma_r$ (price exponent) | 1.5 | 3.0 |
| **Noise Components** | $\sigma_L$ (load noise) | 0.01 | 0.05 |
| | $\sigma_P$ (price noise) | 0.05 | 0.15 |

Chronos consistently improves the performance of vanilla Chronos across all settings—zero-shot, full fine-tuning, and LoRA fine-tuning. Note that the performance of Hopformer variants differs slightly from the main experiment section, as we conducted additional hyperparameter tuning for the GBDT-based covariate regressors (e.g., XGBoost, LightGBM) to ensure a fairer comparison.

Table 4 summarizes the Mean Absolute Scaled Error (MASE) of Hopformer (built on top of Chronos) and vanilla Chronos when the available context ranges from 32 to 512 time steps. Table 5 summarizes the MASE of the models across varying prediction length. Four findings stand out. (i)**Zero-shot performance**: Hopformer consistently beats Chronos at every context length, with the largest gain ($\sim 37.1\%$) at the shortest window of 32 steps, as visualized in Figure 8. This suggests that the covariate-driven expert pool provides valuable signal when historical information is scarce. (ii) **Fine-tuned performance**: After full-parameter or LoRA fine-tuning, Hopformer still yields lower error than Chronos. Removing covariate effects in the first stage appears to simplify the residual dynamics, making the subsequent transformer easier to adapt. (iii) **Cross-sectional aggregation**: In this two-covariate setting, SPA and Lasso deliver comparable accuracy, indicating that with a very small covariate set the sparsity prior in SPA offers little advantage over a standard $\ell_1$-penalised regression. (iv) **Long-horizon prediction**: Hopformer also surpasses Chronos at every horizon, as visualized in Figure 8. The largest gain ($\sim 25.8\%$) locates at the longest horizon of 120 steps, again demonstrating the value of the covariate-driven expert pool as the prediction window expands.

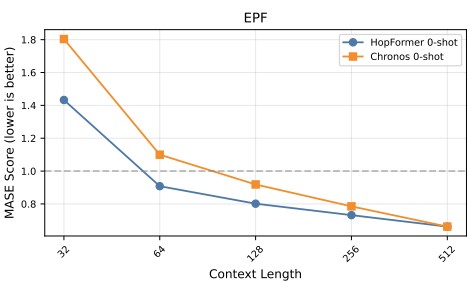 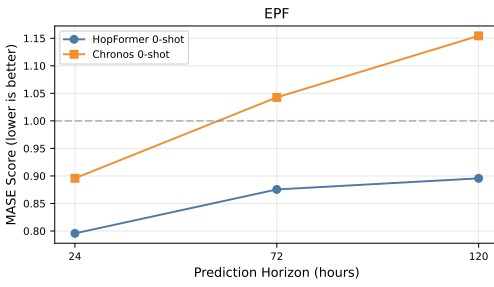

(a) Effect of context length on forecasting.   (b) Effect of prediction horizon on forecasting.

Figure 8: Model robustness across varying context lengths and forecast horizons. Only the informative portion of each figure is shown for clarity.

## C.1 LIMITATION AND DISCUSSION

Hopformer is designed to leverage future covariates; when these are absent its benefit naturally diminishes.

Although Hopformer achieves strong results on most benchmarks, two practical constraints limit the breadth of our evaluation.

**Illness and M5**: On the Illness benchmark (Wang et al., 2024) the series come with seven past covariates but no future ones, so the cross-sectional stage receives little forward-looking signal and Hopformer largely reduces to its residual transformer, yielding only marginal gains over Chronos. The situation is different for the M5 (Makridakis et al., 2022) competition data: rich item-level covariates are available, yet the full dataset contains around 30K hierarchically linked series and several million training points. Training the required gradient-boosted covariate regressors at that scale demands tens of gigabytes of system RAM and many hours of hyper-parameter search, which exceeded the fixed computational budget for this submission. Handling such industry-size datasets will require memory-efficient regressors (e.g., online trees) or sharded training pipelines—an important direction for future work.

**Datasets without covariates**: For univariate or multivariate series (Aksu et al., 2024; Godahewa et al., 2021; Makridakis & Hibon, 2000; Makridakis et al., 2018) that lack future covariates, the cross-sectional stage collapses to a zero-prediction expert, making Hopformer equivalent to its residual transformer sub-model (e.g., Chronos). Because this setting provides no opportunity to test the proposed aggregation mechanism, we omit those results from the main paper and treat pure-series forecasting as an orthogonal problem. Extending Hopformer with automated feature extraction or self-supervised pretraining may restore an advantage in covariate-free domains, and we plan to investigate this in future work.

## D VISUALIZATION OF HOPFORMER INFERENCE

Figure 4 decomposes Hopformer's inference pipeline on the Sale1 benchmark. A clear pattern emerges: once the covariate signal (row 3) including promotional, temperature, and price effect is removed from input time series (row 2), the remaining series (row 3) become markedly smoother and more periodic, with promotional spikes eliminated. This confirms that the expert pool has successfully captured most exogenous variation, leaving the residual module to be a simpler and precisely structured pattern.

This qualitative evidence complements our quantitative results: Hopformer isolates high-variance exogenous effects in the first stage and hands a simpler forecasting task to the residual transformer, leading to the accuracy gains reported in Method Section.

## E ABLATION STUDY: FINE TUNING TIME SERIES FOUNDATIONAL MODELS

Through this ablation study, we provide insight into how finetuning could improve the forecasting performance of two state-of-the-art foundational models. Table 9 compares the effectiveness of zero-shot forecasting with that of full-shot forecasting and of forecasting with a model that was finetuned

using LoRA. All experiments use a context length of 512, a forecast horizon of 24, 100 Monte Carlo samples per forecast, and 20 rolling windows. Note that we performed zero shot forecasting on a computer different from the one used for zero-shot forecasting in the ablation study in Section 5. For LoRA finetuning, we update only the query, key, value, and output projection matrices with rank=8, scaling factor $\alpha = 16$, and a 5 % dropout rate, in 100 gradient steps; full-shot models likewise train for 100 steps on all parameters.

Although the results varied across models and datasets, in most cases, the LoRA-finetuned model matches or even slightly outperforms the full-shot variant. For example, on the EPF dataset, full-shot Chronos attains an MASE of 0.674 and an MAPE of 1.252, while LoRA finetuning nearly matches in performance with MASE = 0.720 and MAPE = 1.265. An instance in which we see LoRA surpassing full-shot finetuning is Moirai on the EPF dataset: LoRA has MASE 0.709 and MAPE 0.108, which outperforms full-shot finetuning (0.849/0.127). These results demonstrate that a lightweight, 100-step LoRA update can recover—and in several cases exceed—the accuracy of full-shot training, though its benefit varies by dataset.

Table 9: Forecasting performance (MASE and MAPE; lower is better) of Chronos and Moirai on the four datasets. Boldface highlights the best-performing metsosohod for each variant.

| Models | Chronos | | | Moirai | | |
|---|---|---|---|---|---|---|
| Variants | 0-shot | Full | LoRA | 0-shot | Full | LoRA |
| EPF MASE | **0.662** | 0.674 | 0.720 | 0.809 | 0.849 | **0.709** |
| EPF MAPE | **1.180** | 1.252 | 1.265 | 0.119 | 0.127 | **0.108** |
| Sale1 MASE | 1.095 | **0.927** | 0.971 | 0.621 | 0.527 | **0.500** |
| Sale1 MAPE | 0.951 | **0.679** | 0.707 | 0.823 | 0.677 | **0.580** |
| Sale2 MASE | 0.542 | 0.307 | **0.301** | 0.566 | **0.545** | 0.557 |
| Sale2 MAPE | 0.518 | 0.315 | **0.309** | 2.440 | **2.222** | 4.452 |
| Elec. MASE | 0.817 | 0.770 | **0.756** | 0.984 | **0.757** | 0.870 |
| Elec. MAPE | 0.083 | 0.079 | **0.078** | **0.402** | 0.407 | 0.442 |

# F  ADDITIONAL MODEL COMPARISON: HOPFORMER VS. CHRONOSX VS. TIMEMIXER

Table 10: Forecasting performance (MASE and MAPE; lower is better) of Hopformer, ChronosX, and TimeMixer. Boldface highlights the two lowest metric values in each row.

| Models | Hopformer | | | ChronosX | | | TimeMixer | |
|---|---|---|---|---|---|---|---|---|
| Variants | 0-shot | Full | LoRA | 0-shot | Full | LoRA | Full | LoRA |
| Sale1 MASE | 0.946 | 0.761 | 0.819 | 1.088 | 0.708 | **0.642** | **0.301** | 0.925 |
| Sale1 MAPE | 0.915 | 0.631 | 0.686 | **0.371** | **0.531** | 0.529 | 1.323 | 2.342 |
| Sale2 MASE | 0.340 | **0.270** | **0.264** | 0.783 | 0.392 | 0.413 | 0.425 | 1.381 |
| Sale2 MAPE | 0.423 | 0.306 | 0.301 | 0.248 | **0.186** | **0.178** | 2.516 | 4.327 |
| Elec. MASE | 0.765 | 0.737 | 0.730 | **0.083** | 0.090 | **0.070** | 0.273 | 1.023 |
| Elec. MAPE | 0.078 | 0.075 | 0.075 | 0.061 | **0.042** | 0.295 | **0.030** | 0.138 |
| Illness MASE | 0.381 | **0.330** | **0.329** | 2.380 | 1.852 | 1.389 | 2.065 | 5.069 |
| Illness MAPE | 0.133 | 0.115 | 0.115 | 0.067 | **0.053** | **0.040** | 0.246 | 0.481 |
| EPF MASE | 0.654 | 0.650 | **0.642** | 1.196 | 1.096 | – | **0.377** | 1.044 |
| EPF MAPE | 1.114 | **1.112** | **1.109** | 6.223 | 6.039 | – | 2.102 | 6.853 |
| M5 MASE | 1.006 | **0.984** | 0.989 | 1.091 | – | – | **0.733** | 1.097 |
| M5 MAPE | 0.703 | **0.670** | **0.589** | 0.697 | – | – | 2.885 | 5.825 |

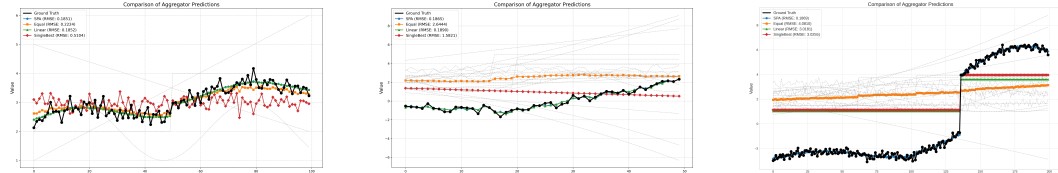

Figure 9: Simulated grocery-sales example (grey curves are individual regressors encoding trend, seasonality, and promotion effects). Left: Linear mixture of 3 regressors over 100 time steps with added noise—SPA matches ordinary least-squares (LR) aggregation. Middle: Non-linear combination of 20 regressors over 50 time steps—SPA outperforms LR by capturing interaction effects. Right: Same non-linear setting over 200 time steps—SPA continues to beat LR, demonstrating robustness as series length grows.

# G    ALABATION STUDY ON SPA

# H    FORECASTING VISUALIZATIONS FROM TABLE 3.

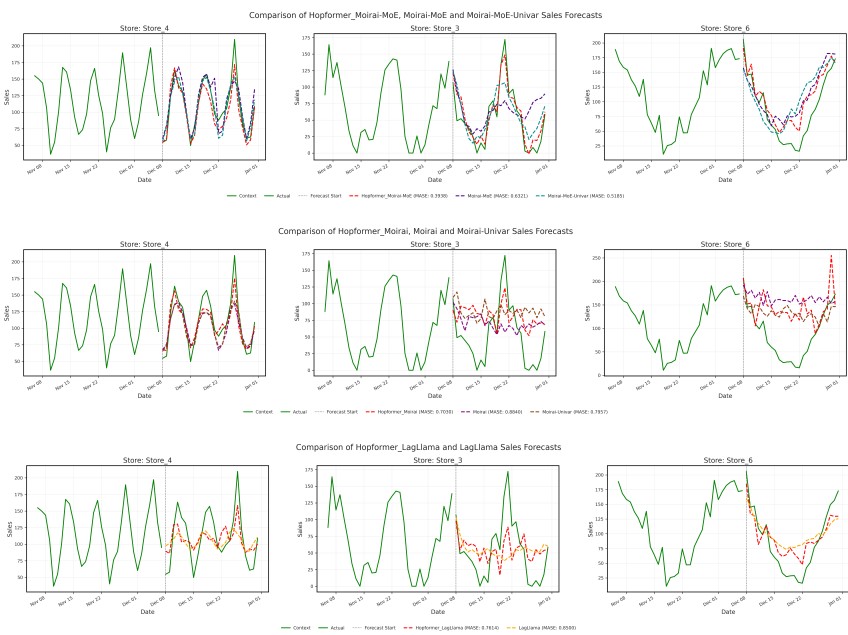

Figure 10: Forecasting results on the Sales1 dataset. top: Moirai-MoE; middle: Moirai; bottom: Lag-Llama.

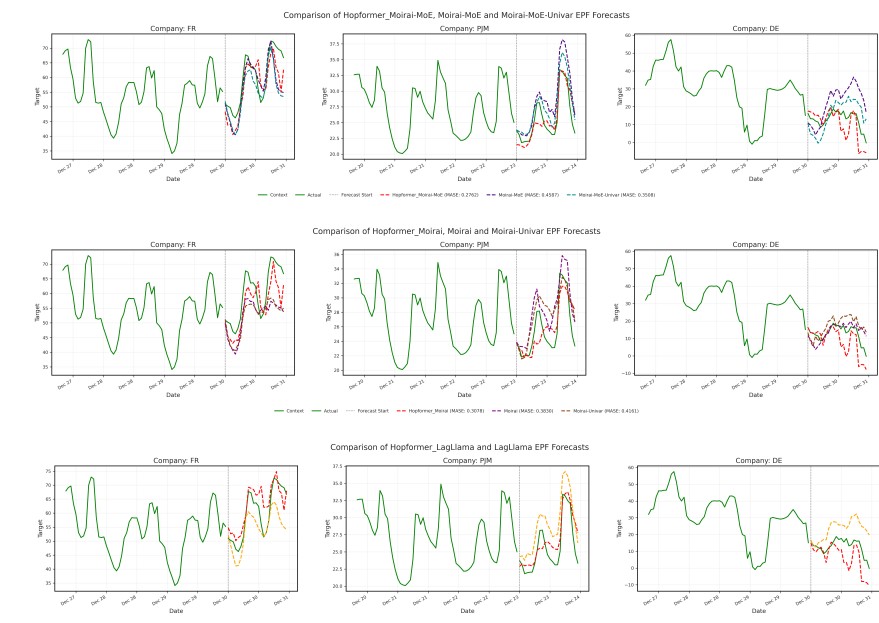

Figure 11: Forecasting results on the `EPF` dataset. top: Moirai-MoE; middle: Moirai; bottom: Lag-Llama.

## I    CODE USAGE

Listing    1:    Minimal    hopformer    usage.    The    interface    mirrors `autogluon.timeseries.Predictor`, enabling effortless integration into existing pipelines.

Here, we desribe the simple usage of Hopformer library [2]. The `Predictor` wrapper exposes Hopformer through the same API as `autogluon.timeseries.Predictor`, so existing Auto-Gluon (Shchur et al., 2023) scripts require only a one–line replacement. Listing 1 trains a Hopformer model on the SALE1 dataset and produces 24-step forecasts. Any AutoGluon regressor (e.g. XGBoost (Chen & Guestrin, 2016), LightGBM (Ke et al., 2017), CatBoost (Prokhorenkova et al., 2018)) can be dropped into the expert pool by editing the `regressor_types` list, and fine-tuning can be toggled with a single Boolean flag.

## J    IMPLEMENTATION DETAILS

We describe the detailed implementation of Hopformer, which operates through two main phases during training and prediction:

**Fitting Phase:** The algorithm first learns to model covariate effects using an ensemble of cross-sectional regressors, then fine-tunes a pre-trained foundation model on the resulting residuals. This sequential approach ensures that the foundation model focuses exclusively on temporal patterns that cannot be explained by covariates.

**Prediction Phase:** For forecasting, the algorithm applies the learned decomposition in reverse: it first extracts residuals from the context data, predicts future residuals using the foundation model, then reconstructs the final forecasts by adding back the predicted covariate effects.

This design enables the framework to leverage both the covariate modeling capabilities of tradi-tional ML and the temporal pattern recognition abilities of foundation models, while avoiding the interference that would occur if both components were learned simultaneously.

---

[2]Data    and    code    are    available    at    https://www.dropbox.com/scl/ fo/q4t08x79w1jxq2tnkz15d/ACXuC5nO6yS17cH696oaP0g?rlkey= e9ogypv287u8232l06dzn0u28&dl=0.

---

**Algorithm 1** Hopformer Fitting

---

**Require:** Training data $\mathcal{D} = \{(y_{i,t}, \mathbf{x}_{i,t}, \mathbf{s}_i) \mid i \in \mathcal{I}, t \in \mathcal{T}_i\}$ where $y_{i,t}$ is target, $\mathbf{x}_{i,t}$ are covariates, $\mathbf{s}_i$ are static features

**Require:** Prediction horizon $H$, context length $L$, model path $\text{path}_{\text{foundation}}$, Lora configuration $\mathbf{\Theta}_{\text{lora}}$

**Ensure:** Fitted model $\mathbf{\Theta} = \{\theta_{\text{scaler}}, \theta_{\text{reg}}, \theta_{\text{foundation}}\}$

 1: **Data Preprocessing**
 2: $\theta_{\text{scaler}} \leftarrow \text{LocalStandardScaler}()$
 3: $\mathcal{D}_{\text{scaled}} \leftarrow \theta_{\text{scaler}}.\text{fit\_transform}(\mathcal{D})$ {Normalize targets per time series}
 4: **Phase 1: Covariate Effect Modeling**
 5: $\theta_{\text{reg}} \leftarrow \text{CrossSectionalRegressor}(\text{models} = \mathcal{M}, \text{hyperparams} = \mathcal{H}_{\text{reg}})$
 6: $\mathcal{R} \leftarrow \theta_{\text{reg}}.\text{fit\_transform}(\mathcal{D}_{\text{scaled}}, L)$ {Get residuals with context length}
 7: **Phase 2: Foundation Model Fine-tuning**
 8: $\text{hyperparams} \leftarrow \{\text{model\_path} : \text{path}_{\text{foundation}}, \text{lora\_config} : \mathbf{\Theta}_{\text{lora}}, \text{horizon} : H, \text{context} : L\}$
 9: $\theta_{\text{foundation}} \leftarrow \text{FoundationModel.fit}(\mathcal{R}, \text{hyperparams})$
10: **return** $\mathbf{\Theta} = \{\theta_{\text{scaler}}, \theta_{\text{reg}}, \theta_{\text{foundation}}\}$

---

**Algorithm 2** Residual Chronos Prediction

---

**Require:** Context data $\mathcal{D}_{\text{context}} = \{(y_{i,t}, \mathbf{x}_{i,t}) \mid i \in \mathcal{I}, t \in \mathcal{T}_{\text{context}}\}$

**Require:** Future covariates $\mathbf{X}_{\text{future}} = \{\mathbf{x}_{i,t} \mid i \in \mathcal{I}, t \in \mathcal{T}_{\text{future}}\}$

**Require:** Static features $\mathbf{S} = \{\mathbf{s}_i \mid i \in \mathcal{I}\}$

**Require:** Context length $L$

**Require:** Fitted model $\mathbf{\Theta} = \{\theta_{\text{scaler}}, \theta_{\text{reg}}, \theta_{\text{foundation}}\}$

**Ensure:** Forecasts $\hat{\mathbf{Y}} = \{\hat{y}_{i,t} \mid i \in \mathcal{I}, t \in \mathcal{T}_{\text{future}}\}$

 1: **Data Preprocessing**
 2: $\mathcal{D}_{\text{scaled}} \leftarrow \theta_{\text{scaler}}.\text{transform}(\mathcal{D}_{\text{context}})$ {Apply learned scaling}
 3: **Phase 1: Extract Residuals and Predict**
 4: $\mathcal{R}_{\text{context}} \leftarrow \theta_{\text{reg}}.\text{transform}(\mathcal{D}_{\text{scaled}}, L)$ {Remove covariate effects with context length}
 5: $\hat{\mathcal{R}}_{\text{future}} \leftarrow \theta_{\text{foundation}}.\text{predict}(\mathcal{R}_{\text{context}}, \mathbf{X}_{\text{future}}, L)$ {Predict residuals with context length}
 6: **Phase 2: Reconstruct Full Predictions**
 7: $\hat{\mathbf{Y}}_{\text{scaled}} \leftarrow \theta_{\text{reg}}.\text{inverse\_transform}(\hat{\mathcal{R}}_{\text{future}}, \mathbf{X}_{\text{future}}, \mathbf{S}, \mathcal{D}_{\text{scaled}})$
 8: **Inverse Scaling**
 9: $\hat{\mathbf{Y}} \leftarrow \theta_{\text{scaler}}.\text{inverse\_transform}(\hat{\mathbf{Y}}_{\text{scaled}})$ {Return to original scale}
10: **return** $\hat{\mathbf{Y}}$ {Final forecasts with quantile predictions}

---

## NOTATION

- $\mathcal{I}$: Set of time series identifiers
- $\mathcal{T}_i$: Time indices for series $i$
- $H$: Prediction horizon length
- $L$: Context length for foundation model
- $\mathcal{Q}$: Set of quantile levels $\{0.1, 0.2, \ldots, 0.9\}$
- $\mathcal{M}$: Set of regression models $\{\text{XGBoost}, \text{RandomForest}, \text{CatBoost}, \ldots\}$
- $\Theta_{\text{lora}}$: LoRA configuration $\{\text{rank } r, \text{alpha}, \text{dropout}, \text{target\_modules}\}$

## MODEL COMPONENTS

- **LocalStandardScaler** ($\theta_{\text{scaler}}$): Per-series z-score normalization
- **CrossSectionalRegressor** ($\theta_{\text{reg}}$): Ensemble of ML models for covariate effects
- **FoundationModel** ($\theta_{\text{foundation}}$): Pre-trained transformer (Chronos/Moirai/LagLlama)

