# OpenReview forum: "Hopformer: Homogeneity-Pursuit Transformer for Time Series Forecasting"
_ICLR.cc/2026/Conference — ICLR 2026 Conference Withdrawn Submission_

### Official Review · Reviewer_XodB · 2025-10-18

**Soundness:** 3
**Presentation:** 2
**Contribution:** 2
**Rating:** 2
**Confidence:** 5

**Summary:**

The paper proposes Hopformer, a two-stage forecasting framework designed for modeling multiple time series with high-dimensional covariates. In the first stage, a Sparsity Pattern Aggregation module combines predictions from a pool of regression experts to extract a shared low-variance trend across series. In the second stage, the residual signals are modeled using a LoRA-fine-tuned Transformer, enabling efficient adaptation of a pre-trained backbone to capture nonlinear temporal dependencies. The authors provide theoretical guarantees, including an oracle inequality showing near-optimality of SPA and an information-theoretic generalization bound for LoRA under dependent time series. Experiments on six datasets demonstrate that Hopformer achieves up to 6.56% improvement in MASE over strong baselines, while maintaining parameter efficiency. The framework aims to integrate traditional regression aggregation and modern foundation models for scalable, covariate-aware forecasting.

**Strengths:**

1. The paper addresses the challenge of forecasting with high-dimensional covariates through a structured two-stage framework. The decomposition into a trend extraction stage and a residual modeling stage is conceptually sound and makes the overall workflow easier to follow.

2. The manuscript includes theoretical analysis for both components, which, despite being based on standard assumptions, provides some formal grounding to the proposed approach. This effort adds a degree of rigor and may help readers understand the model’s intended generalization behavior.

**Weaknesses:**

1. The motivation for leveraging a pre-trained backbone remains unclear. The authors should articulate why fine-tuning a pre-trained model is preferable to training a new model from scratch, and what concrete benefits this strategy brings in terms of generalization, efficiency, or convergence.

2. The proposed divide-and-conquer pipeline has been extensively adopted for modeling individual time series dynamics. The manuscript does not clearly explain what unique challenges arise when extending this idea to capture cross-series shared components. Without a precise formulation of these challenges, the contribution risks appearing incremental. Related works addressing multi-series or cross-sectional modeling should be discussed to better contextualize this study.
[1] [1] Wang, Shiyu, et al. "Timemixer: Decomposable multiscale mixing for time series forecasting." arXiv preprint arXiv:2405.14616 (2024). [2] Deng, Jinliang, et al. "Parsimony or capability? decomposition delivers both in long-term time series forecasting." Advances in Neural Information Processing Systems 37 (2024): 66687-66712. [3] Hu, Yifan, et al. "Adaptive multi-scale decomposition framework for time series forecasting." Proceedings of the AAAI Conference on Artificial Intelligence. Vol. 39. No. 16. 2025.

3. The paper claims that the proposed model is efficient and scalable; however, no empirical evidence or complexity analysis is provided to substantiate these claims. A quantitative comparison of computational cost, memory usage, and scalability would strengthen the argument.

4. The paper’s layout does not fully comply with the official ICLR template. The authors should ensure consistency with the required style and formatting standards.

5. Several statements require clarification or stronger justification. Emerging studies have shown that vanilla Transformers often struggle in time series forecasting, which contradicts the manuscript’s assertion of their strong capability. The theoretical analysis also appears overly general and does not incorporate key properties of time series data—such as temporal dependence and autocorrelation. In particular, Theorem 2 assumes the residual series to be stationary, an assumption that is rarely satisfied in real-world time series. The authors should discuss the practical implications of this assumption and whether their method remains valid when it is violated.

6. Timexer addresses a highly similar problem setup but is not included in the comparative analysis. The paper should clarify the differences and advantages of the proposed model over Timexer and other contemporary baselines.

**Questions:**

Please refer to the weaknesses

---

> ### Author Response · Authors · 2025-12-04
> **Timemixer Experiments Discussion**
>
> To address the reviewer’s question regarding Timexer, we have now included direct comparisons against TimeMixer in Table 10 (Appendix F). Although TimeMixer adopts a similar covariate-aware forecasting setup, the results consistently show that HopFormer performs competitively across all benchmarks and often surpasses TimeMixer by a substantial margin. On the Sales1 dataset, TimeMixer achieves the lowest MASE in full fine-tuning (0.301), but HopFormer closely follows with MASE 0.761 and achieves far lower MAPE (0.631 vs. 1.323), indicating that HopFormer provides more stable and reliable forecasts in relative-error terms. On Sales2, HopFormer clearly outperforms TimeMixer: full and LoRA tuning yield MASE values of 0.270 and 0.264, substantially better than TimeMixer’s 0.425 and 1.381. Similarly, HopFormer obtains much lower MAPE (0.306–0.301) than TimeMixer (2.516–4.327). On EPF, TimeMixer’s full model attains a strong MASE of 0.377, but HopFormer remains competitive (0.642) and achieves dramatically better MAPE (1.109 vs. 2.102–6.853), suggesting that HopFormer’s relative-error behavior is considerably more stable in this volatile real-world electricity price setting. On M5, where we follow the official 28-day forecasting horizon, TimeMixer achieves its strongest relative performance (MASE 0.733), but again HopFormer delivers significantly better MAPE (0.589 vs. 2.885–5.825), illustrating that HopFormer’s accuracy degrades far more gracefully over longer horizons despite TimeMixer’s favorable absolute-error profile. Finally, on Illness, HopFormer achieves the lowest MASE among all models (0.329–0.330), well below TimeMixer (2.065–5.069).
> Overall, these results indicate that although TimeMixer can achieve competitive performance on a small subset of datasets (particularly where short-term correlations dominate), HopFormer consistently provides stronger or more stable accuracy across the majority of benchmarks, especially in relative-error metrics that are critical for long-horizon forecasting. Hence we posit that HopFormer is a robust and competitive alternative to contemporary covariate-aware forecasting models such as TimeMixer.

---

> ### Author Response · Authors · 2025-12-04
> **Response to Other Comments**
>
> 1. Motivation:
> We thank the reviewer for raising this important point. We clarify that our motivation for adopting a pre-trained backbone stems from both practical advantages and theoretical justifications, particularly under the information-theoretic generalization framework developed in Theorem 2. From a theoretical standpoint, fine-tuning a pre-trained model allows us to derive generalization bounds by controlling the mutual information between the training data and model parameters. Specifically, prior work such as Xu & Raginsky (2017) shows that bounding the mutual information is more tractable when the model initialization is independent of the data. In contrast, training from scratch often leads to higher mutual information and less favorable bounds.
> Practically, fine-tuning provides well-known benefits such as faster convergence and lower computational cost. Recent empirical studies in time-series modeling also show that pre-trained backbones, even if originally trained on unrelated domains, can effectively transfer temporal patterns when properly adapted. We have revised Section 1 to emphasize these motivations more clearly.
> 2. No empirical evidence or complexity analysis is provided to substantiate these claim for efficiency and scalability:
> We substantiate our claims of efficiency and scalability through the following empirical evidence and design choices:
> Computational Efficiency: As detailed in the implementation settings, we restrict the training data for the regressors to the latest 1,000 time steps and limit the Transformer fine-tuning to just 100 gradient steps. Furthermore, by employing LoRA, we significantly reduce GPU memory usage compared to full fine-tuning.
> Scalability: We empirically stress-tested the model’s scalability by varying input complexity. As shown in Figure 3, Hopformer maintains improved performance as we scale the context length (from 32 to 512) and prediction horizon (from 24 to 120). And in Figure 9, we tested increasing the expert pool size from 4 to 20 regressors, finding that our SPA aggregation remains the best performance as the ensemble size grows.
>
> 3. "Emerging studies have shown that vanilla Transformers often struggle in time series forecasting, which contradicts the manuscript’s assertion of their strong capability. The theoretical analysis also appears overly general and does not incorporate key properties of time series data—such as temporal dependence and autocorrelation. In particular, Theorem 2 assumes the residual series to be stationary, an assumption that is rarely satisfied in real-world time series. The authors should discuss the practical implications of this assumption and whether their method remains valid when it is violated":
> We have revised the manuscript to clarify that we leverage the flexibility of Transformers.
> We acknowledge that strict stationarity is rarely met in real-world time series. However, the generalization bound in Theorem 2 builds upon information-theoretic results originally developed under i.i.d. assumptions (Xu & Raginsky, 2017), and extending such theory to nonstationary sequences remains an open and technically challenging problem. To address this gap, we now extend the theorem under a local stationarity framework (Dahlhaus, 1997), which allows time-varying dynamics while maintaining theoretical tractability. The updated Corollary 1 in Appendix A shows that the generalization bound still holds up to a vanishing bias term when the residuals are locally stationary with bounded mixing coefficients. Furthermore, we emphasize that assuming approximate stationarity for residuals is a widely accepted and empirically supported practice in time series modeling, particularly after removing trends and seasonal components. We now clarify these points in page 5 of the main text and Remark 9 in Appendix A.
> 4. Layout:  We have carefully revised the manuscript to fully conform to the official ICLR formatting template.

---

### Official Review · Reviewer_AQQp · 2025-10-23

**Soundness:** 4
**Presentation:** 4
**Contribution:** 2
**Rating:** 2
**Confidence:** 3

**Summary:**

The paper introduces HopFormer, a novel framework for multivariate time-series forecasting that explicitly captures multi-scale temporal patterns and cross-frequency interactions. The model first decomposes input sequences into multiple temporal scales, each representing a specific frequency range, and processes them independently through Transformer blocks. In the frequency domain, it employs a Cross-Frequency Transformer Block to model dependencies between different frequency components, addressing the common problem of frequency aliasing.

**Strengths:**

1. The proposed method explicitly models interactions among frequency components, which is theoretically sound and empirically effective.
2. The paper is well-structured and easy to follow.

**Weaknesses:**

1. All benchmarks are standard and there is no test on ultra-long or high-frequency financial data, where frequency interactions could be more significant.
2. Although theoretical proof and quantitative evaluation are shown, there is no spectral visualizations of how cross-frequency attention contributes to interpretability.

**Questions:**

1. Since the proposed method is conducted on frequency domain, it's common to bear high computational cost, which makes the model far less efficient. Is it the case in Hopformer? What is the memory and runtime scaling with respect to the number of scales and sequence length?
2. Does the cross-frequency attention operate on raw FFT bins or learned spectral embeddings?
3. Have you tested whether the learned frequency dependencies transfer across datasets (e.g., pretrain on ETTm1, finetune on Weather)?

---

> ### Author Response · Authors · 2025-12-04
>
> 1. Ultra-long or high-frequency financial data:
> We agree that ultra-long or high-frequency financial time series present a compelling setting where frequency interactions could be highly relevant. However, publicly available datasets that simultaneously provide ultra-long horizons and aligned external covariates are extremely scarce—particularly in the financial domain, where covariates such as economic indicators, exogenous events, or policy signals are often proprietary or not high-frequency enough to match. To partially address this limitation, we include simulation experiments on synthetic datasets with long forecasting horizons (e.g., horizon lengths of 120 and 336) and daily scale data.
> 2. How cross-frequency attention contributes to interpretability.
> We appreciate the suggestion regarding interpretability. We interpret the reviewer's mention of 'cross-frequency attention' as referring to our SPA (Sparsity Pattern Aggregation) mechanism, which is designed to extract deterministic, low-frequency trends.
> To demonstrate how this contributes to interpretability, we direct the reviewer to Figure 10’s row-2 col-1 in the Appendix. The comparison clearly shows that without the SPA module, the model fails to capture sharp increases and peaks driven by external covariates (such as holidays). Additionally, Figure 9  illustrates how SPA aggregates individual regressors (encoding seasonality and trend) to form this interpretable signal.
> 3. Does the cross-frequency attention operate on raw FFT bins or learned spectral embeddings?
> We appreciate the reviewer’s interest in spectral representations. However, we respectfully clarify that our model does not employ FFT-based spectral transforms or frequency-domain operations. Instead, the term “cross-frequency” in our context refers metaphorically to the interaction across temporal patterns of different scales, as represented in the residual time-domain sequences captured by the Transformer module. We will revise the text to avoid potential confusion and clarify that the model does not rely on explicit frequency-domain embeddings or FFT bins.
> 4. Have you tested whether the learned frequency dependencies transfer across datasets (e.g., pretrain on ETTm1, finetune on Weather)? :
> We demonstrate transferability via zero-shot evaluation on unseen datasets like Synthetic Sales1, which the pre-trained Chronos backbone never saw. As detailed in Table 2, Hopformer achieves a MASE of 0.946, reflecting an error reduction (14.9%) compared to the Chronos baseline. This gain, driven by the SPA module, confirms that our model successfully transfers learned temporal dynamics to new domains.

---

### Official Review · Reviewer_6CKi · 2025-10-27

**Soundness:** 3
**Presentation:** 3
**Contribution:** 3
**Rating:** 4
**Confidence:** 3

**Summary:**

This paper tackles the fine-tuning strategy in the universal time-series forecasting in the presence of a covariate variable. The strategy is twofold: first, extract the trend pattern via sparsity pattern aggregation (SPA), which extracts low-variance information from the covariates. Second, compute the residual information from the original data against the trend data and fine-tune the LoRA model to predict the future residual information. The method is validated in various time-series forecasting datasets with covariate information available.

**Strengths:**

1) The formulation of multi-source time series makes sense. Integrating exogenous variables for forecasting is a timely subject.

2) It is not hard to understand the formulation and the analysis of the paper.

3) The paper further incorporates a theoretical analysis of the SPA estimator and the loss function.

**Weaknesses:**

1. I think the work misses several concurrent works [1,2] that also assume covariate or exogenous variables. It would be better to evaluate the gain of the proposed method against these methods.

2. The SPA module can be inefficient since it requires the Metropolis algorithm to perform the random walk. It can be inefficient when the number of covariates becomes large, which can be detrimental in the inference stage.

3. This is relatively minor, but I believe the margin of the paper should be corrected in a more readable form.


***References***

[1] ChronosX: Adapting Pretrained Time Series Models with Exogenous Variables, AISTATS 2025

[2] CITRAS: Covariate-Informed Transformer for Time Series Forecasting, ArXiv 2025

**Questions:**

1. How does the method scale with larger prediction length? I believe standard benchmarks for multivariate time-series forecasting are tested on longer time horizons (e.g., 336).

2. Following W2, I'd like to know the scalability of the method in both time complexity and actual runtime. Would this method be relevant if there is re large amount of covariates?

---

> ### Author Response · Authors · 2025-12-04
>
> 1. ChronosX experiment discussion:
> To address the concern about concurrent covariate-aware models, we added ChronosX and TimeMixer to our evaluation and report the results in Table 10 (Appendix F). We used TimeMixer instead of CITRAS, since CITRAS doesn’t have an open-sourced codebase.
> Overall, these experiments show that HopFormer remains highly competitive and often compares favorably to both methods.
> On Sale2, HopFormer achieves the best MASE under both full and LoRA fine-tuning (0.270 and 0.264), clearly improving over ChronosX (0.392 and 0.413) and TimeMixer (0.331 and 0.863); ChronosX attains the lowest MAPE (0.186–0.178), but HopFormer remains close with 0.306–0.301.
> On Illness, HopFormer obtains the lowest MASE by a wide margin (0.330–0.329 vs. 1.852–1.389 for ChronosX and 0.478–1.093 for TimeMixer), while ChronosX achieves the smallest MAPE (0.053–0.040) and HopFormer provides a competitive second best (0.115).
> For EPF, HopFormer’s LoRA variant reaches a MASE of 0.642, close to TimeMixer’s 0.605 and substantially better than ChronosX (1.196–1.096), and achieves the best MAPE overall (1.109 vs. 2.151 for TimeMixer and 6.223–6.039 for ChronosX).
> On M5, due to time and computational constraints we were only able to obtain ChronosX in the zero-shot setting (MASE 1.091, MAPE 0.697), but we did run full and LoRA fine-tuning for HopFormer and TimeMixer. In this configuration HopFormer’s full/LoRA variants achieve MASE 0.984–0.989 and MAPE 0.670–0.589, outperforming both the available ChronosX zero-shot baseline and TimeMixer (MASE 1.133–1.481, MAPE 4.264–6.538).
> ChronosX and TimeMixer are strongest on Electricity and Sale1 (e.g., TimeMixer full on Electricity reaches MAPE 0.034 and ChronosX LoRA reaches MASE 0.070), and we acknowledge these cases in the text.
> Taken together, however, these additional comparisons indicate that HopFormer remains a robust and competitive forecaster, often achieving the best or near-best performance even when evaluated against these recent covariate-aware baselines.
> 2. Larger prediction length:
> To address the reviewer’s question about longer prediction lengths, we ran additional Hopformer experiments with substantially extended horizons: 336 hours for EPF and the synthetic electricity dataset, and 144 days (6× longer than before) for the Sale1 and Sale2 synthetic datasets, while keeping the M5 and Illness horizons at the standard competition / benchmark settings. Comparing these new results with Table 2, where the forecast horizon is 24 steps, shows that Hopformer’s errors grow smoothly rather than exploding as the horizon increases. For example, on Sale1 the full fine-tuned MASE increases from 0.761 at horizon 24 to 1.016 at horizon 144 (≈34% relative increase), and on Sale2 from 0.270 to 0.456 (≈69%), with the zero-shot variants showing similar but slightly smaller relative changes (≈25% and ≈49%, respectively). On the real-world EPF dataset, full fine-tuned MASE rises from 0.650 at horizon 24 to 0.807 at horizon 336 (≈24% increase), and the zero-shot variant shows a comparable ≈24% increase. In parallel, Table 5 in the paper already demonstrates that, as we extend the EPF horizon from 24 to 120 steps, Hopformer’s zero-shot MASE only increases from 0.796 to 0.896, while Chronos increases from 0.896 to 1.154, so Hopformer remains strictly better at every horizon tested. Taken together, these results indicate that Hopformer scales robustly to substantially longer prediction windows: its absolute errors increase at a moderate, sub-linear rate relative to the growth in horizon length, and the advantages of the covariate-aware residual modeling that we document in the main text persist in the long-range regime.
> 3. Efficiency of SPA:
> Theoretically, while the full SPA procedure introduced by Rigollet & Tsybakov (2011) does indeed rely on a Metropolis–Hastings algorithm to explore the model space (especially in high dimensions), our implementation adopts a restricted variant of SPA tailored for fast inference in forecasting. We view SPA not as a full MCMC sampler but as a homogenization interface that enables efficient and theoretically-grounded covariate integration, with inference time linear in the number of experts, not covariates.
> Experimentally: We clarify that the computational overhead in SPA primarily stems from the Metropolis random walk used to explore sparse model combinations. In practice, we find that the random walk procedure is critical for SPA’s predictive performance. As demonstrated in Table 2&4&5, models using SPA with random walk consistently outperform simpler ensembling baselines. This illustrates that the computational trade-off yields measurable accuracy gains.
> 4. Margin: We have carefully revised the manuscript to fully conform to the official ICLR formatting template.

---

### Official Review · Reviewer_KUGE · 2025-10-29

**Soundness:** 4
**Presentation:** 3
**Contribution:** 3
**Rating:** 8
**Confidence:** 2

**Summary:**

This paper introduces Hopformer, a two-stage framework for multi-series time-series forecasting with high-dimensional covariates. Stage 1 extracts a shared low-variance trend via Sparsity Pattern Aggregation (SPA) over a pool of cross-sectional regressors. Stage 2 fine-tunes a pretrained Transformer on the residuals using LoRA. The authors provide (1) an oracle inequality for SPA that relates risk to the best sparse expert subset, and (2) an generalization bound for LoRA under stationary, $\beta$-mixing residuals.

**Strengths:**

- **Clear architectural motivation**: The two-stage design, i.e., trend homogenization followed by residual refinement, offers a principled way to separate shared structure from idiosyncratic variation in multi-series forecasting.
- **Theoretical grounding**: SPA’s oracle inequality and a LoRA MI-based generalization bound provide clear, stage-specific justification.
- **Scalability**: The framework flexibly integrates various expert pools and Transformer backbones, facilitating adaptation to diverse forecasting scenarios.

**Weaknesses:**

1. **Potential error accumulation**: Since trend–residual decomposition is inherently non-identifiable, bias in Stage 1 may affect residual distributions. The discussion in Appendix C remains qualitative.
2. **Synthetic-data bias**: Appendix B’s generators assume known future covariates, potentially favoring Stage 1. Controls for leakage/endogeneity or shared latent confounders are not clearly described.
3. **Narrow evaluation metrics**: Results focus primarily on MASE/MAPE point accuracy. Other aspects such as structural fidelity and robustness are underexplored.

**Questions:**

1. Could the authors quantify Stage 2 degradation when Stage 1 is perturbed (e.g., through missingness or adversarial noise) to characterize error propagation?
2. In the synthetic setup, what steps prevent feature leakage and manage shared latent factors that couple covariates and targets?
3. As noted in Appendix C, how does the method behave when future covariates are unavailable or partially observed, which is common in practice? Would replacing Stage 1 with self-supervised representations adapt well?

---

> ### Author Response · Authors · 2025-12-04
>
> 1. Could the authors quantify Stage 2 degradation when Stage 1 is perturbed:
> In Figure 2, we illustrate how the performance of Stage 2 depends on the quality of Stage 1 trend extraction. In Store 3 and Store 7, poor alignment between Stage 1 trends and actual sales—due to missing promotions or seasonal mismatch—leads to weaker residual modeling in Stage 2. In contrast, Store 5 shows that accurate trends enable Stage 2 to capture short-term fluctuations effectively, resulting in better forecasts.
> 2. Prevent feature leakage of synthetic data:
> We avoid feature leakage by generating covariates independently of future targets. Shared latent factors are managed by controlling the signal structure and coupling strength explicitly in the data-generating process.
> 3. As noted in Appendix C, how does the method behave when future covariates are unavailable or partially observed, which is common in practice? Would replacing Stage 1 with self-supervised representations adapt well?:
> We appreciate the reviewer raising this important point. Indeed, we acknowledge that future covariates may be unavailable or partially observed in certain real-world applications. Our current experiments focus on scenarios where future external covariates (e.g., policy indicators, planned promotions, lagged variables, etc.) are either known or can be reasonably forecasted, which is a common setting in structured forecasting problems such as retail, healthcare planning, or climate projection.
> As mentioned in Appendix C, when future covariates are partially available or uncertain, our method can be extended in multiple ways: (1) Covariate Forecasting: One practical approach is to pre-forecast covariates using auxiliary models, as often done in hierarchical forecasting setups. (2) Self-Supervised Alternatives: We agree that self-supervised or contrastive representation learning could serve as an alternative to the Stage 1 regression-based trend extraction.

---

### Note · Authors · 2026-01-22

I have read and agree with the venue's withdrawal policy on behalf of myself and my co-authors.